# Glacier evolution in high mountain Asia under stratospheric sulfate aerosol injection geoengineering

Liyun Zhao[1,2], Yi Yang[1], Wei Cheng[1], Duoying Ji[1,2], John C. Moore[1,2,3]

[1]College of Global Change and Earth System Science, Beijing Normal University,

19 Xinjiekou Wai St., Beijing, 100875, China

[2]Joint Center for Global Change Studies, Beijing, 100875, China

[3]Arctic Centre, University of Lapland, P.O. Box 122, 96101 Rovaniemi, Finland

Corresponding author: john.moore.bnu@gmail.com

**Abstract:**

Geoengineering by stratospheric sulfate aerosol injection may help preserve mountain glaciers by reducing summer temperatures. We examine this hypothesis for the glaciers in High Mountain Asia using a glacier mass balance model driven by climate simulations from the Geoengineering Model Intercomparison Project (GeoMIP). The

G3 and G4 schemes specify use of stratospheric sulphate aerosols to reduce the radiative forcing under the Representative Concentration Pathway (RCP) 4.5 scenario for the 50 years between 2020 and 2069, and for a further 20 years after termination of geoengineering. We estimate and compare glaciers volume loss for every glacier in the region using a glacier model based on surface mass balance parameterization under

climate projections from three Earth System Models under G3, five models under G4 and six models under RCP4.5 and RCP8.5. The ensemble projections suggest that glacier shrinkage over the period 2010-2069 are equivalent to sea-level rises of $9.0\pm1.6$ mm (G3), $11.5\pm2.5$ mm (G4 excluding HadGEM2-ES), $15.5\pm2.3$ mm (RCP 4.5) and $18.5\pm1.7$ mm (RCP8.5). Although G3 keeps the average temperature from increasing

in the geoengineering period, G3 only slows glacier shrinkage by about 50% relative to losses from RCP8.5. Approximately 72% of glaciated area remains at 2069 under G3 compared with about 30% for RCP8.5. The termination of geoengineering at 2069 under G3 leads to sudden temperature rise of about $1.3^{\circ}$ C and corresponding increase in annual mean glacier volume loss rate from 0.17% $a^{-1}$ to 1.1% $a^{-1}$, which is higher

than the 0.66% a$^{-1}$ under RCP8.5 during 2070-2089.

**keywords**: sea level rise; mass balance; climate impacts

## 1. Introduction

High Mountain Asia (HMA) contains the largest number of glaciers outside the polar

regions. These glaciers provide water for many large and important rivers (e.g.

Brahmaputra, Ganges, Yellow, Yangtze, Indus, and Mekong), and most, but not all,

have shrunk over recent decades (Yao et al., 2012). The response of these glaciers to

future climate change is a topic of concern especially to the many people who rely on

glacier-fed rivers for purposes such as irrigation.

Glacier evolution is expected to be sensitive to climate change. Temperature and

precipitation are the important climate factors affecting glaciers. Geoengineering is a

method of offsetting the global temperature rise from greenhouse gases, although

inevitably also altering other important climate parameters, such as precipitation and

global atmosphere and ocean circulation teleconnection patterns (Tilmes, et al., 2013;

Ricke, et al., 2010). There have been various studies on mountain glacier change under

future climate scenarios such as A1B, and the various Representative Concentration

Pathway (RCP) scenarios (Marzeion et al., 2012; Radić et al., 2014; Zhao et al., 2014).

In contrast to glaciers in higher latitudes, many on the Tibetan Plateau are summer

accumulation type (e.g. Fujita et al., 2000), that is both surface snow fall and melting

occur overwhelmingly in the 3 summer months of June, July and August, with little

mass gain or loss throughout the remaining 9 months of the year. However some

glaciers, especially in the northwestern parts of HMA are winter accumulation type

(Maussion et al., 2014). Hence, the glaciers are affected by both the South Asian

monsoon system and the westerly cyclonic systems, depending on specific location

across the region, thus the region integrates the climate response to two important global

circulation systems (Mölg et al., 2013).

Glacier responses to geoengineering scenarios has been limited to studies on global

responses based on semi-empirical models (Moore et al., 2010;Irvine et al. 2012) or

from simplified ice sheet responses (Irvine et al. 2009; Applegate et al., 2015) or

implications of climate model (McCusker et al., 2015), with nothing to date on mountain glacier impacts.

In this paper, we predict glacier area and volume change for every glacier in HMA under projections from 6 Earth System Models (ESM) simulations of climate under the Geoengineering Model Intercomparison Project (GeoMIP) G3 and G4 scenarios (Kravitz et al., 2011). These scenarios envisage use of stratospheric sulphate aerosols to reduce the radiative forcing under the RCP 4.5 greenhouse gas scenario during in a 50 year period from 2020 to 2069 followed by sudden cessation of geoengineering to determine the "termination effect" (Jones et al., 2013) but continued RCP4.5 greenhouse gas forcing for a further 20 years. We address two questions here: (1) Would glacier shrinkage and loss in HMA be alleviated under geoengineering by stratospheric sulfate aerosol injection? (2) How would the glaciers respond to the termination of geoengineering?

## 2. Study region and glacier data

The Randolph Glacier Inventory (RGI) database contains outlines of almost all glaciers and ice caps outside the two ice sheets (Arendt et al., 2015). Our study region covers HMA (26–46° N, 65–105° E), which corresponds to the defined regions of Central Asia, South Asia West and South Asia East in the RGI 5.0. According to the RGI 5.0, the study region contains a total of 94,000 glaciers and a glaciered area of about 110,000 km$^2$. The RGI 5.0 data inside China are based on the Second Chinese Glacier Inventory (Guo et al., 2015), which provides glacier outlines from 2006–2010, except for some older outlines from the First Chinese Glacier Inventory where suitable imagery could not be found - mainly in southern and eastern Tibet (the S and E Tibet RGI 5.0 sub-region), most of which were made in the 1970s. The RGI 5.0 data outside China are from the "Glacier Area Mapping for Discharge from the Asian Mountains" (GAMDAM) inventory (Nuimura et al., 2015) and nearly all come from 1999–2003 with images selected as close to the year 2000 as possible. Because the data range from each data source is only a few years, we take three reference years: 1980, 2009, and 2000, as start

dates for our model simulations of glaciers in S and E Tibet, elsewhere in China, and outside China, respectively.

Following previous authors (Nuimura, 2015; Zhao et al., 2016), we use median altitude from RGI 5.0 for each glacier as a proxy for equilibrium line altitude (ELA) in the respective initial years; that is the altitude on the glacier where the local net surface

mass balance (SMB) is zero. We use the Shuttle Radar Topography Mission (SRTM) version 4.1 (void-filled version; Jarvis et al, 2008) digital elevation model with 90 m horizontal resolution to estimate the elevation range spanned by each glacier.

Field measurements on SMB are rare in the HMA due to difficulty of access to the glaciers. Following Zhao et al. (2014), we collate SMB versus altitude measurements

from 13 glaciers (Table 1 and Fig. 1), to set up parameterizations of mass balance with altitude relative to the ELA for all glaciers.

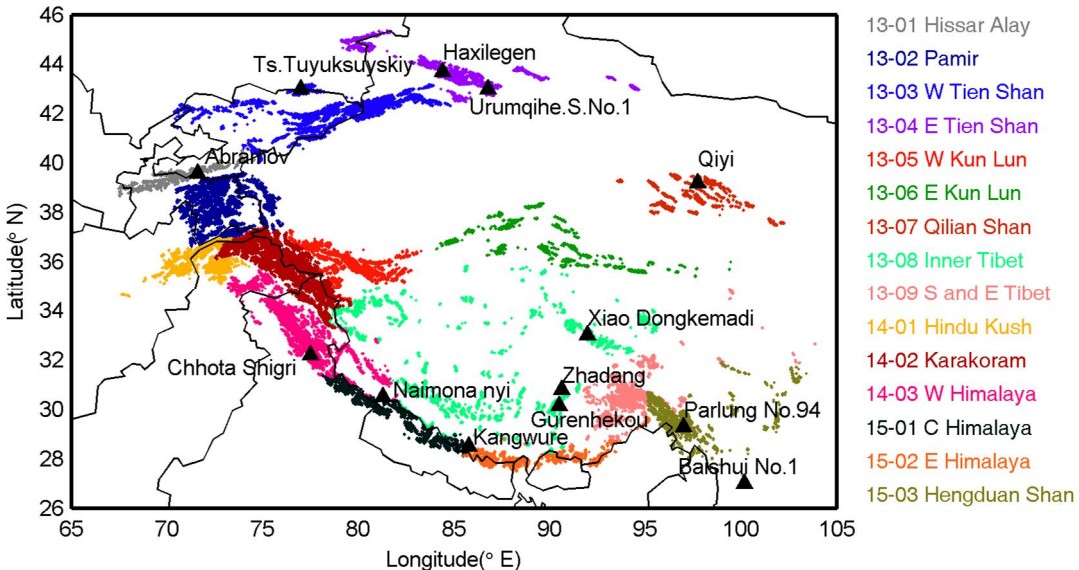

Fig. 1 The HMA region analyzed. Sub-regions of the HMA in RGI 5.0 are listed and colour-coded in the legend. Glaciers with SMB versus altitude measurements (Table 1)

are marked with black triangles.

## 3.  Methodology

### 3.1 Statistical model of glacier change

The statistical model for estimating glacier change is based on Zhao et al (2014; 2016).

Briefly the algorithm can be described as follows. We start from known glacier outlines

from RGI 5.0 and glacier elevation distribution from SRTM 4.1. In the start year, SRTM DEM data (90 m horizontal resolution) inside the glacier outline are interpolated onto a regular grid with a spatial resolution of 10 m covering the glacier surface. Vertical spacing of altitude bands depends on glacier size, taken as 10 m for glaciers with a total elevation difference from top to bottom larger than 100 m, and one tenth of the glacier altitude difference for glaciers with less altitude range.

We parameterize the annual SMB as a function of altitude relative to the ELA for each glacier. We calculate no more than three SMB gradients using in-situ SMB measurements for every glacier in Fig. 1 and Table 1. Following Zhao et al (2014), the SMB–altitude profile is constructed for every glacier by using its own ELA and these SMB gradients. Where SMB data exists in the sub-region, we use them to parameterize the SMB of all glaciers in that sub-region. Otherwise, we use glaciers from nearby sub-regions.

Integrating the SMB over each glacier gives the volume change rate, which is converted to an area change rate using volume–area scaling (Marzeion et al., 2012)

$$dA(n+1) = \frac{1}{\tau_A}\left(\left(\frac{V(n+1)}{c_A}\right)^{1/\gamma} - A(n)\right) \qquad (1)$$

where $A(n)$ is glacier area in the $n_{th}$ year, $V(n+1)$, $dA(n+1)$ are glacier volume and area change rate in the $n+1_{th}$ year, respectively, $c_A = 0.0380 \text{ km}^{3-2\gamma}$ and $\gamma = 1.290$ (Moore et al., 2013), $\tau_A$ is the response time scale of glacier area and calculated as

$$\tau_A(n) = \tau_L(n)\frac{A(n)}{L(n)^2} \qquad (2)$$

where $L(n)$ and $\tau_L$ are glacier length and the response time scale of glacier length in the $n_{th}$ year, respectively. $\tau_L$ is calculated by

$$\tau_L(n) = \frac{V(n)}{A(n) \cdot P^{solid}(n)} \qquad (3)$$

following the scaling of Johannesson et al. (1989) where $V(n)$ and $P^{solid}(n)$ denote glacier volume and the solid precipitation on the glacier in the $n_{th}$ year, respectively. The

initial glacier length $L_{start}$ is estimated by area-length scaling $A_{start} = c_L L_{start}{}^{q_L}$ where $c_L = 0.0180$ km$^{3-q}$ (Radic et al., 2008) and $q = 2.2$ (Bahr et al., 1997). The glacier length change is calculated using the area-length scaling

$$dL(n+1) = \frac{1}{\tau_L} \left( \left( \frac{A(n+1)}{c_L} \right)^{1/q} - L(n) \right). \quad (4)$$

We assume all the area changes take place in the lowest parts of the glacier. The set of glacier surface grid points is updated every year --- the number of the grid points that need to be removed or added is calculated using the area change rate while the elevation of the grid points is updated using SMB.

For retreating glaciers, we remove grid cells near the glacier terminus from the glacier surface grids and get the new glacier terminus position and hence the new outline for the next year. For advancing glaciers, we add grid points to the glacier surface grid, whose elevations are all supposed to be the glacier elevation minimum in the n+1$_{th}$ year, $z_{min}(n+1)$, which is obtained as follows by assuming a constant glacier surface slope,

$$z_{min}(n+1) = z_{max}(n+1) + \frac{L(n+1)}{L(n)} \cdot (z_{min}(n) - z_{max}(n)), \quad (5)$$

where $z_{max}(n+1)$ denotes the glacier elevation maximum in the n+1$_{th}$ year. We also limited the maximal surface increase at any point on the glacier to 15 m above the initial elevation at the starting year. We chose to do this because the valley glacier is physically constrained from growing above the level of the surrounding mountain ridge and side-

walls.

The SMB–altitude profile on each glacier is evolved annually as the ELA changes. And the ELA evolution is estimated by using its sensitivities with respect to temperature and precipitation as follows:

$$ELA_n = ELA_{n-1} + \alpha \Delta T + \beta \Delta P, \quad (6)$$

where $ELA_n$ is the ELA in the $n^{th}$ year from the beginning year, $\Delta T$ and $\Delta P$ are the inter-annual change of summertime (June-July-August) mean air temperature and

annual solid precipitation on the glacier, the coefficients α (unit: m °C$^{-1}$) and β (unit: m m$^{-1}$) are the sensitivity of ELA shift to air temperature change (°C) and precipitation change (m), respectively, which are zonal mean values from energy-balance modelling of glaciers in HMA by Rupper and Roe (2008), see also Zhao et al. (2014).

**3.2 Climate scenarios and downscaling of climate data**

We run the simulations for glacier change from the relevant start years (Section 2) to the year 2089. From the start years to 2013, we use the relatively high resolution, monthly-mean gridded 0.5°×0.5° temperature data from the CRU TS 3.24 dataset (Harris et al., 2014), and 0.5°×0.5° monthly total gridded precipitation data from the Global Precipitation Climatology Centre (GPCC) Total Full V7 dataset (Becker et al., 2013).

For the years 2014 to 2089 we use 4 kinds of climate forcing: experiment RCP4.5, RCP8.5, and results from two GeoMIP scenarios (G3 and G4; Kravitz et al., 2011) which use stratospheric aerosols to reduce the incoming shortwave while applying the RCP4.5 greenhouse gas forcing. In G3 and G4, stratospheric geoengineering of sulphate aerosol injection starts in the year 2020 and ends in the year 2069. In the 50 years of geoengineering under G3 there is close to a balance between reduction of incoming shortwave radiation and the increase in greenhouse gas forcing, while G4 specifies continuous injection of SO$_2$ into the equatorial lower stratosphere at a rate of 5 Tg per year from 2020. The across model spread of temperatures under G4 is larger than under e.g. RCP4.5 (there are too few ensemble member models under G3 to see this) because of differences in how the aerosol forcing is handled, and each model has a different temperature response to the combined long and shortwave forcing (Yu et al., 2015). Following the abrupt end of geoengineering, both G3 and G4 specify 20 years of further simulation from 2070 to 2089.

We derived climate forcing data from three climate models participating in G3, 5 models in G4, six models in RCP4.5 and RCP8.5 (Table 3). We use the Coupled Model Intercomparison Project Phase 5 (CMIP5; Taylor, et al., 2012) output of all models. Yu et al. (2015) noted there was no significant change in surface temperatures after sulphate aerosol was injected in the GISS-E2-R model possibly due to the efficacy of SO$_2$

forcing being relatively small as compared to $CO_2$ forcing in the model. Neither do we also find a termination effect in GISS-E2-R under G3. Therefore, we not use any results from GISS-E2-R. We also exclude the model CISRO-Mk3L due to its very coarse spatial resolution of about $4^o$ and the absence of simulation results in the year 2020 under G4; the models used and their resolutions are listed in Table 3.

Compared with the size of most glaciers in HMA (typically km scale), the CRU, GPCC and climate model grids have rather coarse resolution (Table 3). The direct use of coarse grid points naturally results in a poor representation of the local climate. Hence we downscale both the CRU gridded temperature data, the GPCC gridded precipitation data and the climate model output to a grid based on a land surface topography having resolution of $0.1126^o \times 0.1126^o$ using an altitude temperature lapse rate of $0.65^{\circ}\text{C}/100$ m, an altitude precipitation lapse rate of 3%/100 m, and elevation difference of the fine local grid point relative to the climate model grid.

We bias correct the downscaled model temperatures and precipitation output by using CRU gridded temperature data and GPCC gridded precipitation data as a reference climate. Downscaled series were produced for each climate model for the period 2013 to 2089 under each climate scenario by averaged monthly differences over the baseline period 1980 to 2005 taken from the models' CMIP5 *historical* simulations. We only use summer (JJA) mean near-surface air temperature. Therefore, future temperature time series $T_i(\text{t})$ on each grid point were calculated by

$$T_i(\text{t}) = T_{i,c}(\text{t}) + (\overline{T}_{i,CRU} - \overline{T}_{i,c,history}), \quad \text{i}=6,7,8 \quad (7)$$

where $T_{i,c}(\text{t})$ is monthly mean temperature for the *i*th month from the climate model output from $t = 2013$ to 2089, $\overline{T}_{i,CRU}$ and $\overline{T}_{i,c,history}$ are mean temperature from CRU TS V 3.24 dataset and climate model output, respectively, for the *i*th month averaged over the baseline period 1980-2005 on each grid point.

Future precipitation time series $P_i(\text{t})$ on each grid point were calculated by

$$P_i(\text{t}) = P_{i,c}(\text{t}) \cdot \frac{\overline{P}_{i,GPCC}}{\overline{P}_{i,c,history}}, \quad \text{i}=1,...,12 \quad (8)$$

where $P_{i,c}(t)$ is monthly precipitation for the $i$th month from the climate model output from $t = 2013$ to $2089$, $\overline{P}_{i,GPCC}$ and $\overline{P}_{i,c,history}$ are monthly precipitation from GPCC dataset and climate model output, respectively, for the $i$th month averaged over the baseline period 1980-2005 on each grid point.

The temperature and precipitation on each glacier were calculated by an altitude temperature lapse rate of 0.65℃/100 m, precipitation lapse rate of 3%/100 m, and the elevation difference of the glacier surface elevation relative to the nearest fine grid point. Moreover, the solid precipitation on the glacier is calculated by the fraction of solid precipitation, $f_{solid}$, based on the monthly mean temperature $T_a$ on the glacier as (Fujita and Nuimura, 2011)

$$f_{solid} = \begin{cases} 1, & \text{if } T_a \leq 0^oC \\ 1-\dfrac{T_a}{4}, & \text{if } 0 < T_a < 4^oC \\ 0, & \text{if } T_a \geq 4^oC \end{cases} . \qquad (9)$$

**3.3 Validation of the glacier model and methodology**

In this section we justify the selection of various parameter values used in the method here. In section 5 we indicate how elements in the model and climate forcing affect the uncertainties of the results we produce in section 4, and how those results compare with previous estimates of glacier evolution in HMA.

A crucial parameterization concerns the SMB-altitude gradients. The field data (Table 1) include three more glaciers than those used in Zhao et al. (2014; 2016), and include a benchmark glacier from almost every sub-region. With so few glacier observations available, there is an issue of how representative they are of the general population. For inner Tibet, there are three glaciers (Zhadang, Gurenhekou and Xiao Dongkemadi Glacier) with SMB observations, and they have almost the same SMB-altitude gradients, 0.0041 m m$^{-1}$, over their common elevation range (5515~5750 m, Table 1); two glaciers (Naimona'nyi and Kangwure) in central Himalaya have SMB gradients of 0.0038 m m$^{-1}$ in their common altitude range of 5700~6100 m. These similarities suggest that the measured glaciers share some important characteristics with

the vast majority which are not surveyed.

Next we consider the choices for the initial value of ELA at the start year, different V-A scaling parameters and different ELA sensitivities to summer mean temperature and annual precipitation.

In choosing the initial ELAs for each glacier, there are several reasonable alternatives
(Zhao et al., 2016): i) using ELAs interpolated from the first Chinese glacier inventory, ii) median elevations from RGI dataset, iii) the elevation of the 60th percentile of the cumulative area above the glacier terminus. These three choices lead to a range of about 2.5 mm of global sea level in glacier volume loss at 2050. In this study, we use median elevations from RGI dataset, which corresponds to the median result.

Zhao et al. (2014) showed that different volume-area scaling parameterizations can lead to $\pm 5\%$ range of glacier volume loss. The set of parameters we use in this study corresponds to the lower bound of estimated volume loss, but one that is best matched to the observational dataset of 230 separate glaciers (Moore et al., 2013).

For the ELA sensitivity to summer mean temperature and annual precipitation, we
use the zonal mean values from energy-balance modelling of glaciers in HMA by Rupper and Roe (2008). Alternatively, it can be estimated using an empirical formula for ablation and a degree-day method (Zhao et al., 2016). Zhao et al. (2016) calculated the ELA for nine glaciers in China, India and Kyrgyzstan, and compared them with the observed ELA time series by similarities of decadal trends and also annual variability.
The Rupper and Roe ELA parameterization produced the best fits to observed ELA decadal trends on 9 glaciers, with a correlation coefficient of 0.6 which is significant ($p<0.05$, the values we give for p are single tailed Pearson correlation tests).

Combining the above uncertainties would require a Monte Carlo simulation since the parameters combine non-linearly to produce glacier volume and area change; this is
prohibitively expensive to perform given that a single simulation of all glaciers in HMA requires about 60 cpu hours on an 8 cores computer with parallel computing in Matlab. We did estimate elevation changes for individual glaciers directly from simulated volume and area changes, then calculated the average rate of elevation change for all the glaciers in each sub-region and compared them with remote-sensing estimates from

2003 to 2009 from Gardner and others (2013), Table 2. The correlation coefficient between the Gardner et al. (2013) estimates for the 6 RGI 5.0 sub-regions with data regional and our modeled regional averages is 0.7 which is marginally significant, (p<0.1).

In our simulations we have used constant lapse rates for temperature (0.65℃/100

m) and precipitation (3%/100 m). To check how reliable this is we chose 5 meteorological stations close to glaciers and calculated correlation coefficients for JJA temperature and annual precipitation at the station and at the nearest downscaled grid point from 1980 to 2013 (n=34). Precipitation correlations were higher than 0.85 for all the stations (p<0.001), while temperatures correlations were 0.47-0.85 (p<0.01).

Finally we explored the sensitivity to the choice of dataset used to correct model bias in temperatures. In addition to using historical temperature from the CRU dataset, we also did the simulation using temperature from Berkeley Earth project (1°× 1° resolution; Rohde et al., 2013; http://berkeleyearth.org/data/). That simulation was done using temperature alone as the glacier driver, so precipitation for each glacier was

constant over time. The simulated climate ensemble mean forced volume losses in the period 2010-2069 were +4% (G3), -9% (G4), -11% (RCP4.5) and -13% (RCP8.5) different from the results using the CRU dataset.

## 4.  Results
### 4.1 Climate and glacier change across HMA
### 4.1.1 Temperature and precipitation over HMA

We construct the climate forcing by using CRU temperature data and GPCC precipitation data before 2013 and climate models (Table 3) with model bias correction (Section 3.2) under RCP4.5 from 2014 to 2019, and that under climate scenarios G3,

G4, RCP4.5 and RCP8.5 from the year 2020 to 2089. The JJA mean temperature projections in the whole region under G3, G4, RCP4.5 and RCP8.5 from 2020 to 2089 are shown in Fig. 2. Figure 3 shows the time series of JJA mean temperature and annual precipitation forcing from the beginning years to 2089, with the across-model range from the ensemble members; ranges found are slightly smaller than the regional spread

found by Yu et al. (2015) due to grid-point by grid-point bias correction we apply here.

The multi-model mean temperature under G4 is higher than that under G3 in the geoengineering period. In contrast with the ensemble mean temperature, the HMA mean temperature projected by HadGEM2-ES under G4 is cooler than that under G3; and its G4 temperature is lower than the ensemble mean while its G3 is higher than the

ensemble mean (Fig. 2). The across-model spread in temperature response to G4 is larger than that under G3. Temperatures projected by BNU-ESM are lower than ensemble mean under both G3 and G4.

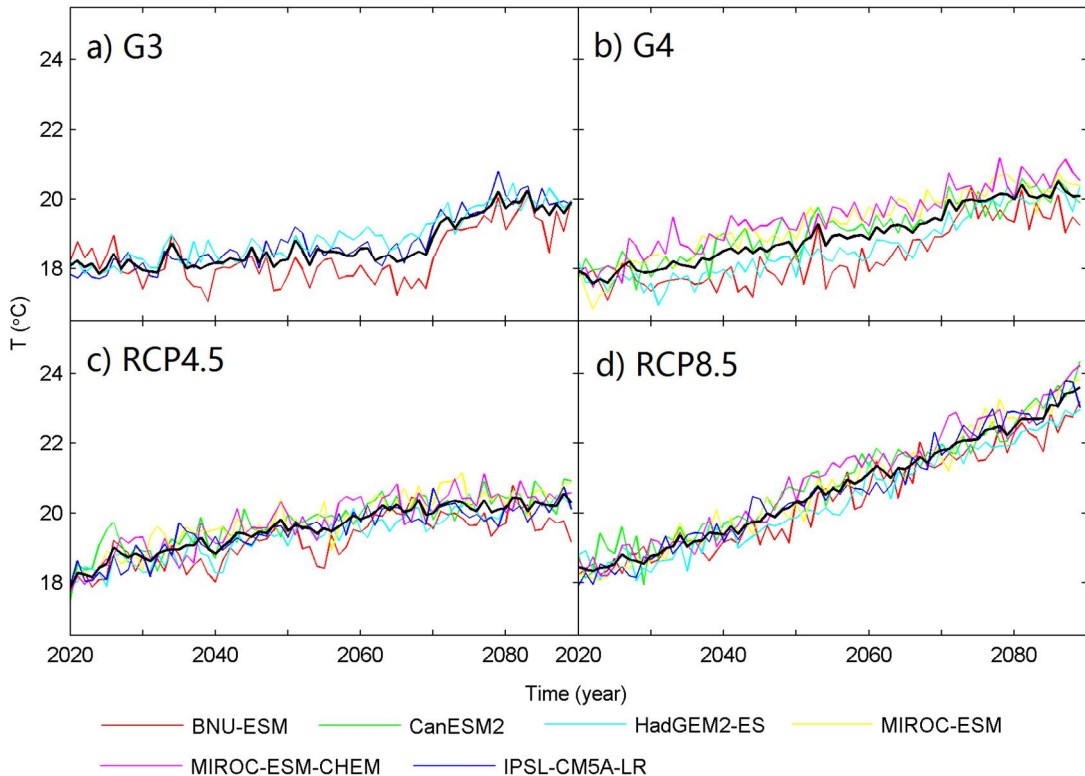

Fig. 2. Time series of summer mean temperature averaged over the downscaled grid

in the whole HMA region projected by ensemble members after model bias correction under climate scenarios G3 (a), G4 (b), RCP4.5 (c) and RCP8.5 (d). Black curve in each plot is the mean of the relevant ensemble (Table 3).

The temperature averages over the whole region and the glaciated parts have similar

trends. Temperatures under RCP8.5, as expected, increase at the highest linear rate among all the scenarios. Temperature rises under RCP4.5 are next highest, and its rate

becomes smaller after about the year 2050 as specified greenhouse gas emissions decline. There are relative coolings of $1.05^{\circ}$ C under G3 and $0.76^{\circ}$ C compared with RCP4.5 during 2020-2069 across the whole region (Fig. 3). Yu et al. (2015) noted that G3 produced a relative cooling under G3 of $0.58^{\circ}$ C and G4 of $0.53^{\circ}$ C in globally averaged temperature over the 2030-2069 period.

There is no trend in temperature under G3 over the geoengineering period (2020-2069). But after the termination in the year 2069, there is a temperature rise of about $1.3^{\circ}$ C over the period 2070-2089 relative to the period 2050-2069 under G3. There is a less obvious termination rise of temperature under G4 than that under G3. This is due to G4 having a constant stratospheric aerosol injection rate of 5 Tg SO2 per year, while G3 gradual ramps-up the aerosol so that about twice as much is needed by 2069, depending upon the sensitivity of the particular model to stratospheric sulphate aerosols. Hence, the radiative impact of terminating G3 is about twice as large as terminating G4, and the termination temperature signal is much more obvious in G3 than G4.

The annual precipitation averages over the whole region do not show obvious trends in any climate scenarios (Fig. 3c). However, the annual solid precipitation averages over the glaciers show decreasing trends in all the scenarios (Fig. 3d) until 2070, which is due to increases in surface air temperature (Fig. 3b). Under RCP8.5, annual solid precipitation averaged over each glacier decreases fastest (2.2 mm a$^{-1}$). Decreases are similar (about 1.5 mm a$^{-1}$) under RCP4.5 and G4 and least (0.3 mm a$^{-1}$) under G3 during the geoengineering period (2020-2069). After the year 2070, there are no trends in annual solid precipitation under G3, G4 and RCP4.5 (Fig. 3d) due to stable temperatures (Fig. 3b).

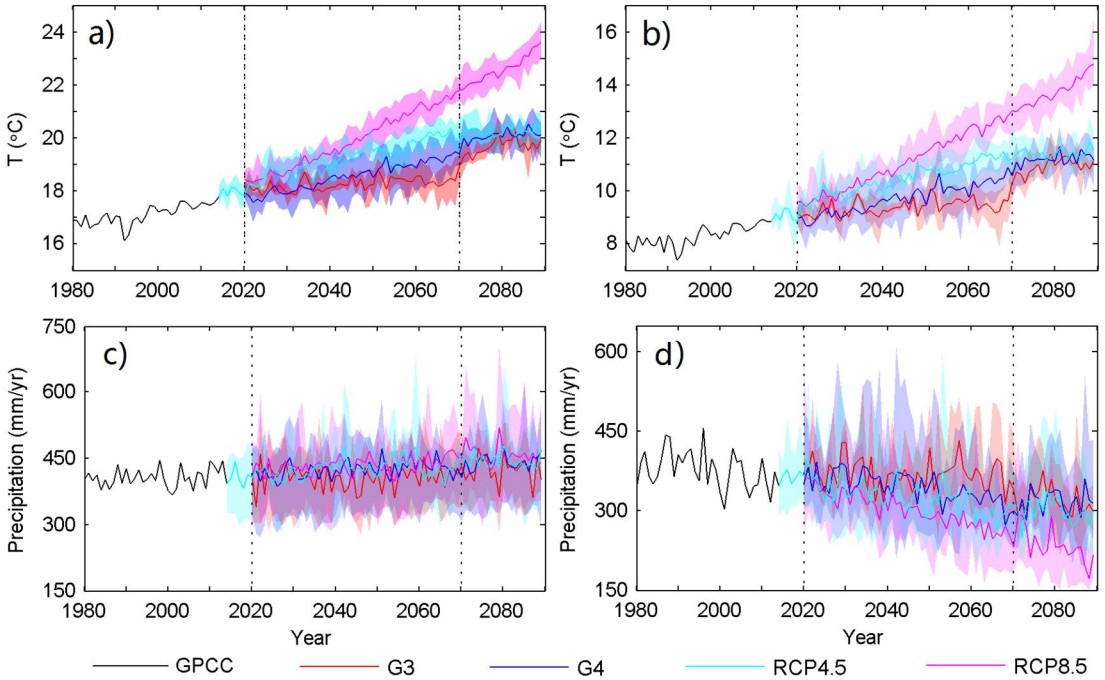

Fig. 3. Time series of summer mean temperature (a, b) and annual precipitation (c, d) averaged over the downscaled grid (section 3.2) in the whole region (a, c) and only in the glaciated region (b, d). Note the different temperature ranges in plot (a) and plot (b) and precipitation ranges in plot (c) and plot (d). Precipitation in plot (d) is the average

annual solid precipitation at the ELA of each glacier in the start year of simulations which is taken here to be representative of each glacier. The solid curves and shadings from 2013 to 2089 are ensemble mean and the across-model spread between ensemble members for each scenario, which are colour-coded in the legend.

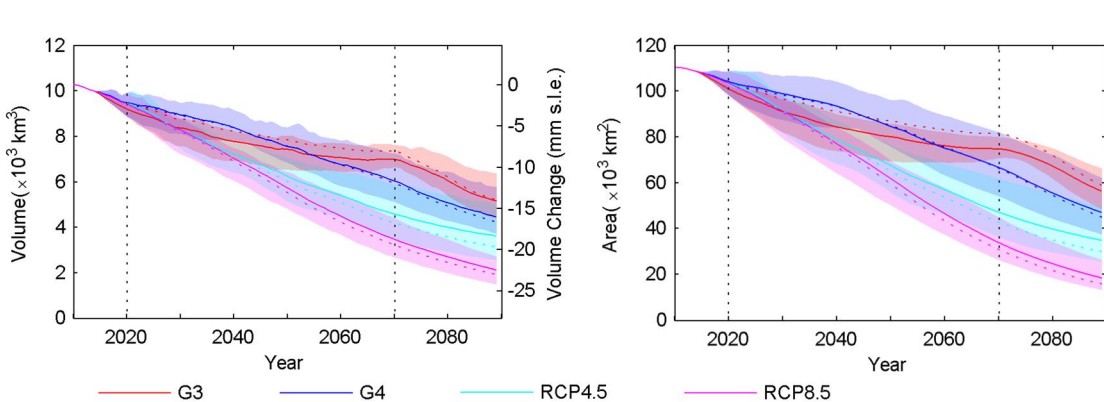

Fig. 4. Total glacier volume in HMA (a) and the equivalent sea level rise assuming an ice density of 900 kg m$^{-3}$ and ocean area of $362 \times 10^{12}$ m$^2$ and area (b) from 2010 to 2089. The solid curves and shadings are means of individual climate model forced

simulations and the across-model spread, for colour-coded scenarios. The dashed curves are results using multi-model ensemble mean temperature and precipitation forcing under each scenario.

### 4.1.2 Glacier changes across HMA

Glacier volume changes for all the glaciers in the study region computed using temperature and precipitation data from the four scenarios are shown in Fig. 4a. The uncertainty we plot is due only to the differences between climate forcing across the models, it does not reflect uncertainty of the glacier model parameters. Volume loss rates increase from G3, G4, RCP4.5 to RCP8.5 for the period 2020-2069. The RCP4.5 and RCP8.5 scenarios produce similar continuous mass loss until approximately 2035 (Fig. 4a) mainly due to the similarity of temperatures projected by RCP4.5 and RCP8.5 in the period 2020-2035 (Fig. 3a), and both show relatively slower loss rates after about the year 2050 probably because the most sensitive glaciers have already retreated before 2050. Volume loss using the climate projected by HadGEM2-ES under G4 is far less than that by other models (Table 4), so we exclude it when calculating the G4 model mean. The multi-model mean glacier volume loss in equivalent to sea-level rise for the whole study region from 2010 to the end of geoengineering in 2069 is 9.0 mm (G3), 11.7 mm (G4), 15.5mm (RCP 4.5) and 18.5 mm (RCP8.5), with 91.8%, 96.0%, 98.5% and 99.7% glaciers retreating under these scenarios (Table 4). These numbers may also be compared with the simulations run using the ensemble mean climate forcing (last row in Table 4), which are all close to the means of the individual model driven mass losses, as are the time varying loss rates (Fig. 4). This is despite the mean climate ensemble including the HadGEM2-ES results for G4. Therefore, the geoengineering schemes G3 and G4 help to reduce glacier mass loss in our simulations, and G3 reduces glacier loss more than G4, which is due to stronger temperature cooling effect under G3 (section 4.1.1).

There is a clear increase in volume loss rate under G3 after 2069 when geoengineering is terminated. Comparing the last 15 years of geoengineering (2055-2069) with the first 15 years of post-geoengineering (2070-2084) shows annual mean

volume loss rate for all the glaciers of 0.17% a$^{-1}$ (referenced to the volume in the year

2010) increases to 1.11% a$^{-1}$, which is higher than the rates of 0.54% a$^{-1}$ for RCP4.5

and 0.66% a$^{-1}$ for RCP8.5. However, the volume loss rate under G4 shows negligible

termination effect; annual mean volume loss rates change from 0.73% a$^{-1}$ to 0.86% a$^{-1}$

before and after the termination. The glacier volume loss over the post geoengineering

period of 2070-2089 for both G3 and G4 are higher than for either RCP 4.5 or RCP8.5

(Table 4). However, by 2070 under both RCP scenarios there is much less glacier ice

volume remaining than under G4, or especially G3. Comparing ice loss rates at

comparable total volumes, loss rates with RCP8.5 are similar to those of post

geoengineering G3.

    As may be expected, glacier area change trends under each climate scenario are

similar to the volume change trends (Fig.4b). We project 53%, 41%, 27% and 14% of

the area in 2010 remaining in the year 2089 under the G3, G4, RCP 4.5 and RCP8.5

scenarios, respectively.

**4.2 Sub-regional climate and glacier changes**

**4.2.1 Sub-regional temperature and precipitation change**

There are three RGI 5.0 regions in HMA: Central Asia, South Asia West and South Asia

East. They are named as Region 13, 14 and 15 and sub-divided into smaller sub-regions

in the RGI 5.0 dataset (Fig. 1). In this section we plot the averages of JJA-mean

temperatures (Fig. 5) and that of annual solid precipitation at the ELA of every glacier

in the start year (Fig. 6) in every sub-region under all the climate scenarios.

    Temperatures under RCP8.5 increase at the highest rates (0.053~0.087$^{\circ}$ C a$^{-1}$)

among all the scenarios, with temperature increases under RCP4.5 in the range of

0.030~0.059$^{\circ}$ C a$^{-1}$ with its rate decreasing after about the year 2050 as specified

greenhouse gas emissions decline. The temperatures rises of 0.030~0.050$^{\circ}$ C a$^{-1}$ occur

under G4 across the sub-regions. Under RCP4.5, RCP8.5 and G4, temperatures increase

slowest in the southeast of the study area (S and E Tibet, C Himalaya, E Himalaya and

Hengduan Shan) and fastest in the northwest (mainly Tien Shan, Hissar Alay,

Karakoram, Pamir, and Hindu Kush).

    There is no trend in temperature under G3 in the geoengineering period (2020-2069)

in all the sub-regions. The temperature cooling projected by G3 compared with RCP4.5 during 2020-2069 is about 1.0 $^{\circ}$C in sub-regions of Central Asia, 1.2 $^{\circ}$C in South Asia West, and 0.8 $^{\circ}$C in South Asia East (Fig. 5). After the termination in the year 2069, there are temperature rises of about 1.07~1.65 $^{\circ}$C over the period 2070-2089 relative to the period 2050-2069 under G3. The post-termination temperatures increase least (about 0.020 $^{\circ}$C a$^{-1}$) in Karakoram and the most (about 0.046 $^{\circ}$C a$^{-1}$) in Eastern Kunlun. The temperature cooling projected by G4 compared with RCP4.5 during 2020-2069 is very similar across all the sub-regions, 0.68~0.86$^{\circ}$C.

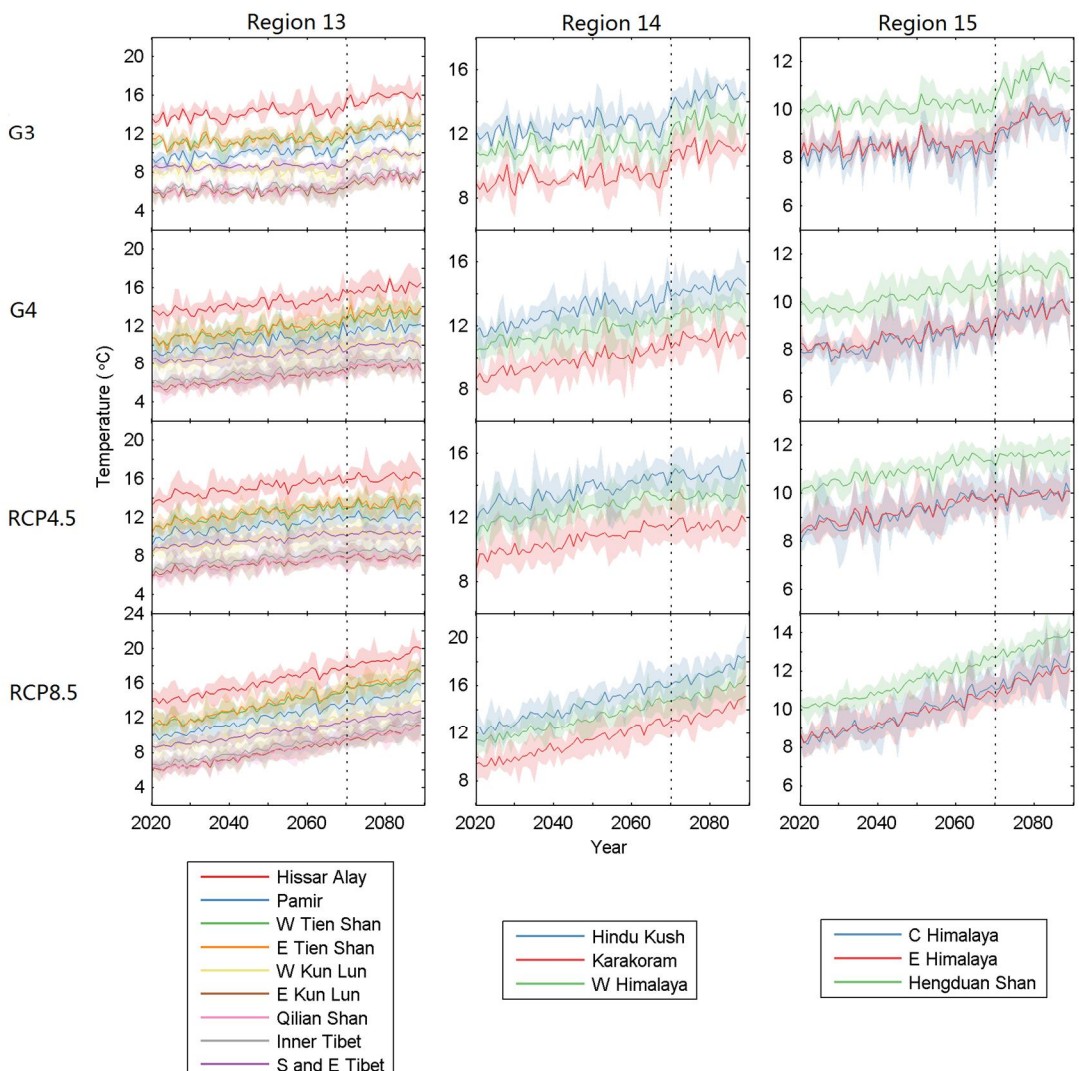

Fig. 5. JJA-mean surface air temperature time series from 2010 to 2089 in the sub-regions of Region 13 (left column), 14 (middle column) and 15 (right column) under scenarios by row from G3 (top), G4, RCP4.5, to RCP8.5 (bottom). Note the different temperature ranges in the panels. The curves and shadings are ensemble mean and the

spread between ensemble members for sub-regions, which are colour-coded in the legend.

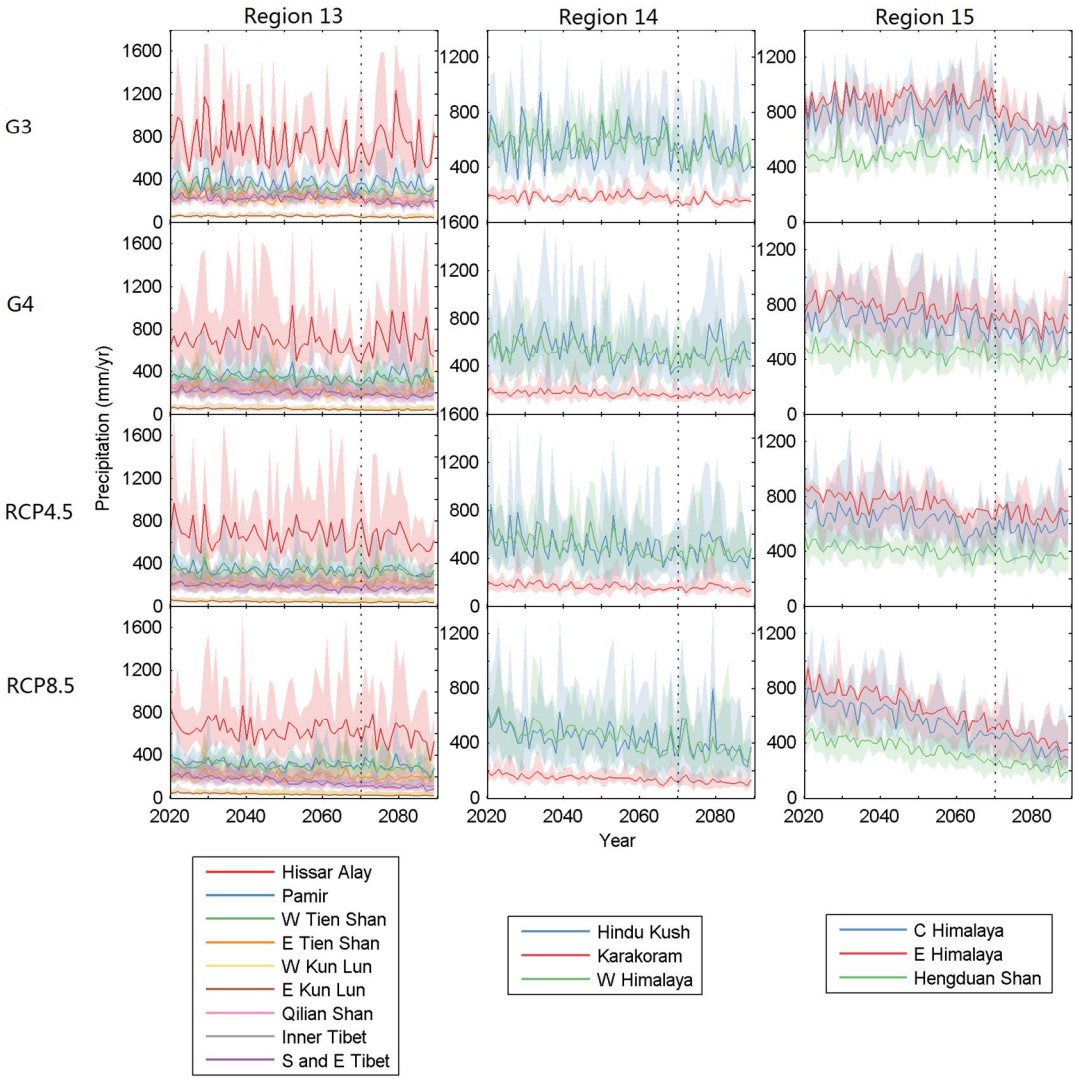

Fig. 6. As that for Fig. 5 but for annual solid precipitation on the glacier.

The annual solid precipitation projected by RCP8.5, and to a lesser degree by RCP4.5 and G4, decreases in all the sub-regions, with the rates larger than 1.5 mm a$^{-1}$ in S and E Tibet, Hindu Kush, W Himalaya and the whole of Region 15 (C Himalaya, E Himalaya, Hengduan). There is no obvious trend of solid precipitation projected by G3 in the geoengineering period (2020-2069) in most sub-regions. But after the geoengineering termination under G3 in the year 2069, there is a significant decrease of solid precipitation in S and E Tibet, Hindu Kush, and the whole of Region 15.

**4.2.2 Sub-regional glacier changes**

Glacier volume changes in the HMA sub-regions are shown in Fig. 7. Glacier volumes in all the sub-regions decrease during the period 2020-2089, with the highest rates under RCP8.5 and the second high rates under RCP4.5, as expected. Glacier volumes decrease with lower rates under G3 and G4 in all the sub-regions except S and E Tibet, inner Tibet, and Hengduan Shan, where glacier volumes increase from the year 2020 to about 2040 under G4, and to the end of geoengineering period under G3 (Fig. 7). The glacier volume triples in S and E Tibet and increases by about 56% in inner Tibet, while increasing slightly in Hengduan Shan in the geoengineering period under G3. The "termination effect" of geoengineering under G3 is significant in most sub-regions.

There are some noticeable difference between means of individual climate model forced simulations and the results using multi-model ensemble mean climate forcing. (Fig. 7). For example, S and E Tibet under all the scenarios, Karakoram under G3, and inner Tibet under G4. This could be because i) individual model differences in temperature and precipitation forcings are large between ensemble members and their means (especially for the 3 model ensemble in G3) in particular sub-regions; ii) glacier hypsometry differences between regions lead to sensitivity under some combinations of forcing when the ELA change is located around large amounts of ice; iii) glacier data inside S and E Tibet was measured in 1970s (section 2) and contains outlines of glacier complexes rather than individual glaciers, which has an impact on the volume estimate because of the non-linearity of volume-area scaling relationship.

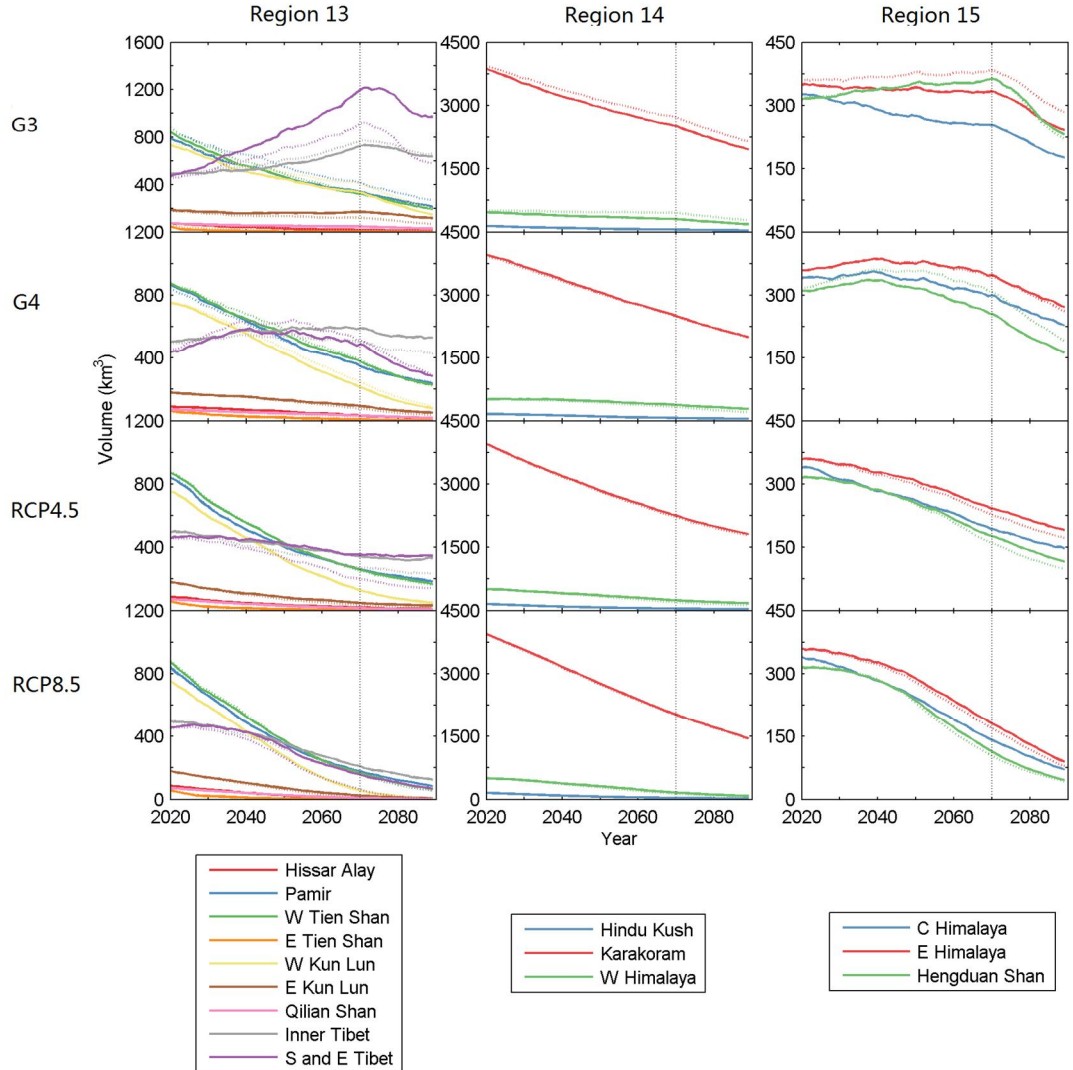

Fig. 7. As that for Fig. 5 but for glacier volume (unit: km³). The solid curves are means of individual climate model forced simulations. The dashed curves are results using multi-model ensemble mean temperature and precipitation forcing under each scenario. The across-model spread for each sub-region is not shown for clarity. Note the difference of glacier volume ranges in the panels.

## 5 Uncertainties in projections

Glacier model parameter selection was discussed in section 3.3, and in more depth by Zhao et al. (2016). In this section we address, and try to estimate, how systematic errors in climate forcing or glacier model parameters cause errors in projections of HMA glacier contributions of sea level rise.

### 5.1 Climate forcing

There are several uncertainties in climate model forcing used to drive the glacier model in this study. The models are also relatively coarsely gridded, certainly compared with the vast majority of glaciers, and so differences may be expected between statistically downscaled forcings based on lapse rates that we use here and that produced from high resolution dynamic climate model forcing.

Firstly, only 3 ESMs participated in G3 but 5 in G4 simply because doing the G3 experiment is difficult and time-consuming to set-up. So the ensemble climate projection by G3 is less robust than that by G4. In many cases it seems that the results from G3 and G4 are statistically similar enough to be combined (Yu et al., 2015, Moore et al., 2015). We tested the differences between RCP8.5, RCP4.5 and G4 using the 4 models in common (Table 4), and find the glacier responses are significantly different ($p < 0.05$). Although there are too few models in common between G3 and G4, the dominant influence of summer melting to the mass balance across the region (Zhao et al., 2016), and the clear difference in temperature across HMA between G3 and G4 (Figs. 2,3) suggest the glacier response in HMA is different between G3 and G4.

Secondly, although the goal of geoengineering schemes is to mitigate temperature rises, it inevitably also alters other important climate parameters, such as precipitation. Simulating change in the Asian monsoon is difficult for climate models under geoengineering since the deep convection involved may also be influenced by chemistry changes in the stratosphere caused by the injected aerosols – most of the ESM models in our study do not have sophisticated aerosol chemistry schemes (though the MIROC-ESM-CHEM model does). Tilmes et al. (2013) showed that changes under the G1 scenario (which specified a much larger shortwave radiation reduction than g3 or G4), produced a weakening of the Asian monsoon and the hydrological cycle by about 5%. The reductions in solid precipitation (Fig. 3) under RCP8.5 are about 1/3 relative to historical levels, and the regions most affected in Region 15 (Fig. 6) are some of those most influenced by monsoon precipitation patterns (Fig. 1). Hence the temperature impact is probably more significant than changes in monsoon precipitation suggested by the G1 results discussed by Tilmes et al. (2013).

Thirdly, we note that the distribution of meteorological stations in the study region

is very sparse, especially in the northwest of this region (Liu and Chen, 2000). Therefore, both the CRU gridded data and data from models projections that we used in this study may have low accuracy for specific glacier regions. This has also implications for the use of very high resolution dynamic models; one such model simulated air temperatures and down-welling radiative fluxes well, but not wind speed and precipitation, producing unstable results when used with the CLM45 land model that simulated ground temperatures and snow cover (Luo et al., 2013). Explicit glacier atmospheric mass balance modelling (Mölg et al., 2013), a technique based on very high spatial and temporal resolution climate data (hourly and 60 m) was used on Zhadang glacier (Fig. 1, Table 1) with in-situ observations available, but not across the general expanse of the glaciated region; this study also noted the importance of wind speed to glacier mass balance in the region influenced by the Indian monsoon. Maussion et al., (2013) demonstrate that 10 km resolution dynamic modelling of the region can be done successfully, and potentially can improve the precipitation modelling over the statistical downscaling methodology we employ here, though to date this is a reanalysis dataset with no prognostic simulations. Zhao et al. (2014 and 2016) used a 25 km resolution regional climate model RegCM3 to drive their simulations of glacier response to scenario A1B. By 2050 under A1B (which is intermediate between RCP4.5 and 8.5 in temperature rises), a sea level rise equivalent to 9.2 mm was projected from HMA. In comparison, our estimates are 11.1-12.5 mm for RCP4.5 and RCP8.5 (Fig. 4).

**5.2 Glacier model**

The model we use is not particularly sophisticated, it simply relies on statistical relationships between mass balance and ELA. Compared with the method used in our previous study Zhao et al. (2014, 2016), we improved our method here by considering the area response time in the volume-area scaling (Eqn. (1)) which is more physical. We also allow the glacier area to grow (section 3.1), giving better estimates of glacier area for advancing glaciers. The motivation to use a relatively simple model must be that it simulates the glaciers well given the available data. As previously discussed there is a shortage of observational data both on glaciers and from climate stations across

HMA. In Section 3.3 we discussed how the model performs when tested against by the limited data available from satellites and ground measurements, in this section we compare the model against previous simulations of HMA glaciers under climate warming, and how its weaknesses may affect the reliability of projected mass changes.

Perhaps a strong limitation on the glacier simulation under geoengineering in our model is the lack of response to the changes in short wave forcing that would be produced under aerosol injection schemes. van de Berg et al. (2011) showed that Greenland mass balance during the Eemian interglacial could not be explained purely by temperature rises but must also include losses due to changes in the shortwave

radiation flux on the ice sheet.

Testing our results for the greenhouse gas scenarios against previous studies; we project glacier volume loss, in equivalent sea-level rise, for all the glaciers from 2010 to 2089 as 18.2±2.5 mm and 22.4±1.3 mm under the RCP 4.5 and RCP8.5 scenarios, respectively. The volume change of all the glaciers in HMA over the 21th century

estimated by Radić et al. (2014) is about 15±5 mm under RCP4.5 and 22±5 mm under RCP8.5. Marzeion et al. (2012) estimate about 15.4±4.5 mm under RCP4.5, and 18.8±4.0 mm under RCP8.5 using projected temperature and precipitation anomalies from an ensemble of 15 CMIP5 climate models. The results projected by our method have higher means but smaller uncertainties than theirs, but do not differ significantly.

The across-model uncertainties we plot here (Fig. 4) are smaller than glacier method uncertainties (section 3.3; Zhao et al., 2016). Hence, more mass balance and meterological stations on glaciers across the region, or longer and higher spatial resolution time series of glacier elevation changes, would better constrain the projected mass losses than simply increasing the number, or resolution, of climate models used

in the simulations. That is the range of mass projections given by the mass balance model with different, but reasonable, choices of data-limited quantities such as the ELA-sensitivity to temperature or the SMB-altitude gradients, is larger than the across model range for each climate scenario.

**6 Summary and Conclusion**

We estimate and compare glaciers volume loss for glaciers in HMA using a statistical model based on glacier SMB parameterization to the year 2089. We construct temperature and precipitation forcing by using CRU temperature data and GPCC precipitation data before 2013, and projections from 6 Earth System Models running RCP4.5 and RCP8.5 and the stratospheric sulphate aerosol injection geoengineering scenarios G3 and G4 with model bias correction and downscaling to a high resolution spatial grid based on fixed altitudinal lapse rates for temperature and precipitation. In assessing how glaciers respond to geoengineering climates, we consider only across-climate model differences between the scenarios rather than uncertainties in glacier mass caused by errors in the glacier model we use. The projections suggest that glacier shrinkage at the end of the geoengineering period in 2069 are equivalent to sea-level rises of 9.0±1.6 mm (G3), 11.5±2.5 mm (G4 excluding HadGEM2-ES), 15.5±2.3 mm (RCP 4.5) and 18.5±1.7 mm (RCP8.5) relative to their volumes in 2010 (Table 4), with 91.8%, 96.0%, 98.5% and 99.7% glaciers retreating under these scenarios. There are clear increases in temperature and glacier volume loss rate under G3 after 2069 when geoengineering is terminated, which is higher than the rate under RCP8.5. But the termination effect under G4 is negligible. Glacier volumes decrease in most sub-regions under all the scenarios, while increase in inner Tibet, S and E Tibet and Hengduan Shan from the year 2020 to about 2040 under G4, and to the end of geoengineering period under G3.

Although G3 keeps the average temperature from increasing in the geoengineering period, G3 only slows glacier shrinkage by about 50% relative to losses from RCP8.5. Approximately 72% of glaciated area remains at 2069 under G3 compared with about 30% for RCP8.5. The reason for the G3 losses is likely to be that the glaciers in HMA are not in equilibrium with present day climate, so simply stabilizing temperatures at early 21$^{st}$ century levels does not preserve them. To do that would require significant cooling, perhaps back to early 20$^{th}$ century levels. Achieving that cooling by sulphate aerosol injection may not be possible. The 5 Tg of $SO_2$ per year specified in G4 is about the same loading as a 1991 Mount Pinatubo volcanic eruption every 4 yr (Bluth et al., 1992). G3 requires increasing rates of injection, to 9.8 Tg for the BNU-ESM at 2069.

As aerosol loading increases, its efficacy decreases as particles coalesce and fall out of the stratosphere faster, while also becoming radiatively less effective (Niemeier and Timmreck, 2015). This effect is so strong that it appears unfeasible to use sulphate aerosols to completely eliminate warming from scenarios such as RCP8.5. Greenhouse gas emissions would require very drastic reduction from present levels, and net negative emissions within the next few decades, to limit global temperature rises to 1.5 or 2℃ (Rogelj et al., 2015). If such targets were met, then it is conceivable that plausible quantities of sulphate aerosol geoengineering may be able to maintain 2020 temperatures throughout the 21st century. Even if this politically very difficult combination of drastic emission cuts and quite aggressive sulphate aerosol geoengineering were done, then our simulations suggest the disappearance of about 1/3 of the glaciated area in HMA by 2069 still cannot be avoided.

**Acknowledgements**

We thanks 2 anonymous referees for very constructive critiques of the paper, all participants of the Geoengineering Model Intercomparison Project and their model development teams, CLIVAR/WCRP Working Group on Coupled Modeling for endorsing GeoMIP, and the scientists managing the Earth System Grid data nodes who have assisted with making GeoMIP output available This study is supported by National Key Science Program for Global Change Research (2015CB953601), and National Natural Science Foundation of China (Nos. 41530748, 41506212, 40905047).

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

Table 1. The benchmark glaciers, their RGI 5.0 sub-regions, and their exact location
(Fig. 1), altitude range, averaged SMB gradients (unit: m m$^{-1}$) is specific altitude

intervals, ELA and SMB data sources.

| Glacier name and Sub-region | Location | Altitude range (m) | Averaged SMB gradients | Period of SMB measurements | Reference |
|---|---|---|---|---|---|
| Abramov Glacier (13-01) | (39º38′N, 71º36′E) | 3600-4700 | 0, z>ELA+200; 0.0088, z<ELA+200. ELA varies in 4050-4450 | 1987-1997 | Glacier mass balance bulletin No. 1-6. |
| Ts. Tuyuksuyskiy Glacier (13-03) | (43º03′N, 77º05′E) | 3400-4200 | 0, z>ELA+100; 0.0057, z<ELA+100. ELA varies in 3600-4200 | 1987-2011 | Glacier mass balance bulletin No. 1-12. |
| Urumqihe S. No.1 Glacier (13-04) (East branch) | (43º06′N, 86º49′E) | 3700-4300 | 0.002, ELA<z<4300; 0.01, z<ELA. ELA varies in 3950~4175 | 1987-2011 | Glacier mass balance bulletin No. 1-12. |
| Haxilegen No.51 Glacier (13-04) | 84°24'E, 43°43'N | 3475-3700 | 0.012 | 1999-2005 | Zhang et al. (2015) |
| Qiyi Glacier (13-07) | (39º14′N, 97º45′E) | 4310-5145 | 0.0042, 4800<z<ELA; 0.0014, z<4800. where ELA=5012 | 2002 Jun-Sep; 2002-03; 2010 | Pu et al. (2005); Wang et al.(2011) |
| Zhadang Glacier (13-08) | (30º28′N, 90º38′E) | 5515-6090 | 0.0041 | 2005-06; 2009 Jun-Jul; 2009 Sep-2010 May; 2010 Aug-Sep | Zhou et al.(2007), Mölg et al. (2012). Yu et al. (2013) |
| Gurenhekou Glacier (13-08) | (30°11'N,90°27'E) | 5550-6020 | 0.0041 | 2004-08 | Yu et al. (2013) |
| Xiao Dongkemadi Glacier (13-08) | (33º04′N, 92º05′E) | 5380-5926 | 0.007, z<ELA; 0.004, ELA<z<5750 where ELA~=5515 | 2008-12 | Zhang et al. 2013) |
| Chhota Shigri Glacier (14-03) | (32º12′N, 77º30′E) | 4000-5600 | 0.003, ELA<z<5600; 0.01, ELA-150<z<ELA; 0.005, 4000<z<ELA-150 where ELA varies in 4855-5180 | Annual average SMB during 2002-10; 2003-04; 2004-05 | Azam et al. (2012); Wagnon et al. (2007) |

| Naimona'nyi Glacier (15-01) | (30º27′N, 81º20′E) | 5600-6150 | 0.0006, z>ELA; 0.0038, 5700<z<ELA; where ELA~=6100 | 2005-2010 | Yao et al. (2012) |
| Kangwure Glacier (15-01) | (28º28′N, 85º49′E) | 5700-6100 | 0.0038, 5700<z<6100; | 2005-2010 | Yao et al. (2012) |
| Parlung No.94 Glacier (15-03) | (29°20'N, 97°0'E) | 5067-5334 | 0.01 | 2006-10 | Yang et al. (2013) |
| Baishui No.1 Glacier (15-03) | 26°59′−27° 17′N, 100°04′−10 0°15′E | 4300-5000 | 0.003,z>ELA 0.01, ELA-250<z <ELA; 0.0035, 4300<z<4650 where ELA =4972 | 2008-09 | Du et al. (2013) |

Table 2 The average rate of elevation change (m a$^{-1}$) for all the glaciers in sub-regions compared with remote-sensing estimates from 2003 to 2009 from Gardner and others (2013).

| Sub-regions | Gardner and others (2013) | Modelled |
|---|---|---|
| Hissar Alay and Pamir | -0.13±0.22 | -0.02±0.49 |
| S and E Tibet | -0.30±0.13 | -0.39±0.75 |
| Hindu Kush and Karakoram | -0.12±0.15 | -0.08±0.29 |
| W Himalaya | -0.53±0.13 | 0.32±0.29 |
| C Himalaya | -0.44±0.20 | -0.62±0.63 |
| E Himalaya | -0.89±0.18 | -1.51±0.59 |
| All HMA | -0.27±0.17 | -0.13±0.60 |

Table 3 climate models and datasets used in this study.

| Name | Reference | Resolution | Data sets |
|---|---|---|---|
| CRU | Harris et al., 2014 | 0.5°× 0.5° | Surface temperature 1980-2013 |
| GPCC | Becker et al., 2013 | 0.5°× 0.5° | Precipitation 1980-2013 |
| BNU-ESM | Ji et al., 2014 | 2.8°× 2.8° | G3,G4, RCP4.5, RCP8.5 |
| CanESM2 | Arora et al., 2011 | 2.8°× 2.8° | G4, RCP4.5, RCP8.5 |
| HadGEM2-ES | Collins et al., 2011 | 1°× 1.9° | G3,G4, RCP4.5, RCP8.5 |
| IPSL-CM5A-LR | Dufresne et al., 2013 | 1.9°× 3.8° | G3, RCP4.5, RCP8.5 |
| MIROC-ESM | Watanabe et al., 2011 | 2.8°× 2.8° | G4, RCP4.5, RCP8.5 |
| MIROC-ESM-CHEM | Watanabe et al., 2011 | 2.8°× 2.8° | G4, RCP4.5, RCP8.5 |

830

Table 4. The volume loss in mm sea-level equivalent, projected using forcing from all the climate models in the period 2010-2069 and 2070-2089 post-geoengineering period under G3, G4, RCP4.5 and RCP8.5. The means of volumes lost driven by individual

model forcing and its standard deviation are shown in the penultimate row. The simulated volume loss using the climate model ensemble mean forcing of temperature and precipitation is shown in the last row. The volume loss is calculated by assuming ice density of 900 kg m$^{-3}$ and ocean area of $362 \times 10^{12}$ m$^2$.

| Scenarios | G3 | | G4 | | RCP4.5 | | RCP8.5 | |
|---|---|---|---|---|---|---|---|---|
| Period / Model | 2010-69 | 2070-89 | 2010-69 | 2070-89 | 2010-69 | 2070-89 | 2010-69 | 2070-89 |
| BNU-ESM | 10.2 | 5.3 | 11.0 | 5.5 | 18.5 | 2.5 | 20.8 | 3.2 |
| CanESM2 | ---- | ---- | 8.3 | 4.1 | 14.0 | 2.0 | 17.8 | 3.5 |
| HadGEM2-ES | 7.2 | 3.4 | 3.2[£] | 3.7[£] | 12.0 | 2.5 | 15.9 | 4.7 |
| IPSL-CM5A-LR | 9.8 | 6.3 | ----- | ---- | 16.7 | 3.2 | 19.5 | 3.8 |
| MIROC-ESM | ----- | ----- | 12.6 | 4.0 | 15.8 | 3.0 | 19.0 | 3.9 |
| MIROC-ESM-CHEM | ----- | ----- | 14.0 | 3.8 | 16.0 | 2.9 | 19.1 | 3.1 |
| Mean ± std | 9.0±1.6 | 5.4±1.0 | 11.5±2.5 | 4.4±0.8 | 15.5±2.3 | 2.7±0.4 | 18.5±1.7 | 3.7±0.6 |
| Ensemble mean climate forcing | 8.1 | 5.9 | 11.7 | 4.7 | 16.6 | 2.9 | 19.2 | 3.6 |

[£]HadGEM2-ES is excluded from the mean of the models in G4