# Peer review of "Glacier evolution in high mountain Asia under stratospheric sulfate aerosol injection geoengineering"

_Atmospheric Chemistry and Physics, 2016_

## Referee Comment (RC1) · Anonymous Referee #1 · 17 Oct 2016

General comments

This paper presents the first analysis of the Himalayan glacier response to solar geo-engineering, analyzing the response to a number of scenarios of future greenhouse gas emissions and stratospheric aerosol geoengineering. The authors employ a mixed empirical / statistical model of glacier change driven by temperature change alone and use statistical down-scaling of the GeoMIP multi-model ensemble to produce the input data for their model. They find that solar geoengineering could slow the rate of retreat of Himalayan glaciers and that this benefit would be lost were solar geoengineering to be terminated.

This study is a novel contribution to the geoengineering literature and addresses an important issue; however, there are a number of shortcomings in the study that need

to be addressed.

Most importantly, the method employed by the authors excludes precipitation which is the most critical climate difference between scenarios which include solar geoengineering and those which don't. Studies have reported substantial reductions in monsoon precipitation in scenarios which include solar geoengineering and so it seems odd to exclude this from the analysis. The authors note that excluding precipitation will likely mean that their estimate of the efficacy of solar geoengineering is an overestimate but it is not clear the magnitude of this overestimate; is it of order 1%, 10%, 100%? I'd strongly recommend that the authors either reintroduce precipitation back into their methodology (the model described can use precipitation but it was excluded) even if this is only to provide a robust sensitivity estimate that justifies its exclusion, and I'd suggest doing this even if there are serious shortcomings in the down-scaling method for precipitation or other methodological concerns. Failing that, a much more detailed justification for excluding precipitation should be made along with some robust regional estimates of the significance that the changes in precipitation should be expected to have. The authors note that Rupper and Roe (2008) calculated zonal-mean sensitivities of SMB to temperature and precipitation; I suggest using these to estimate the sensitivity. This study should make clear what the research needs are to make a better estimate of the effects of solar geoengineering on Himalayan glaciers and the significance of including precipitation is a critical question that this paper does not address well.

I had a number of questions about the validity of the methodology adopted in this study that either require the authors to change the methodology, present some additional justification for their approach, or provide some sense of the sensitivity of their approach to these issues. I list the most significant ones here but there are other minor concerns listed afterwards.

First, the authors make comparisons between the responses of a changing ensemble of models for different experiments. It is unclear how much of the difference between

these experiment-ensemble means is due to the changing make-up of the ensembles and how much due to robust differences in response. This is particularly concerning in the case of G4 were the models produced very different temperature responses due to their differing treatments of the 5Tg of injected SO2.

Second, the authors don't do enough to validate their approach. A comparison against historical data or a more explicit reference to studies which validate this approach is needed. Does this model predict stable, growing or shrinking glacier mass for pre-industrial climate conditions? Is this model in agreement with other similar models?

Third, the authors need to fix parameters for the thousands of glaciers in their study area but only have 13 glaciers to base these parameters on. As they are making a regional analysis they use only the 3 nearest glaciers to fix parameters in a regional manner. It is not immediately clear how different these 13 glaciers are from the results presented and so it is not clear now sensitive the results are to this sampling. The authors don't discuss whether one could expect the sampled glaciers to be representative of their region or why one would expect robust regional differences between the glaciers' behavior.

Finally, the authors downscale temperature to a high-resolution grid however they don't make any assessment of whether this downscaled temperature data matches observations. I'd expect that there could be quite large differences between the real glacier altitude and the altitude of the nearest high-resolution gridpoint given the complex topography. I was also wondering given that the observed altitude of the glaciers is used to calculate the ELA, why not use it directly to perform this lapse-rate adjustment?

Relatedly, I'd recommend that the authors broaden the uncertainties section into a proper discussion that covers the shortcomings of the methodology employed and discusses the sensitivity of the findings to various assumptions. A key question which should motivate the material in this section is – how would the glacier response in a model with a full surface mass balance treatment driven directly by high-resolution

climate data differ?

I had a few concerns about the structure and the focus of the paper that relate to the regional climate analysis. First, it seems strange to separate the regional climate analysis from the regional glacier analysis. Why not start with the whole-region analysis of climate and glacier response and then conduct a regional climate and glacier analysis afterwards? Second, Figures 2 and 3 seem of very little use as there are no cues on these plots to relate the results presented to the regional glacier analysis in figure 7. Given glacier response is calculated across the region or over sub-regions, I'd recommend doing a sub-regional-mean analysis as in figure 7 instead of including figures 2 and 3. This would also help to make clearer how significant excluding precipitation from the analysis was.

Finally, the conclusion needs to be revised as it presents a lot of new results and material, some of which are not appropriate for a conclusion.

Specific comments

L18-21 – unclear sentence: 5 under G4 what?

L23-25 – I don't recall this point being made in the paper.

L30-31 – This is a far-reaching general statement extrapolated from a specific scenario. It holds for the specific case but not for the general case.

L55 – delete hence

L55-57 – These are very different types of reasons and I don't see how the second matters.

L57-60 – Irvine et al. 2012 is a semi-empirical study and McCusker et al. discussed ice-sheet implications of their climate model results but did not simulate ice-sheet or sea-level response at all.

L75 – "regions of newly defined regions of" – fix

[Figure]

L86-89 – This sentence is unclear. Why is this simple? What issues could this pose?

L90-95 – This approach needs some justification and citations – why is it appropriate? Is this a widely accepted approach? Etc.

L98-99 – What precisely is done? I believe this is explained in the next section, if so make that link explicit.

Fig 1 + other lat-lon plots – It would help to make sense of the lat-lon plots if there was some frame of reference beyond the lat-lon coordinates. Would it be possible to mark the numbered regions (referred to later) with boxes or outlines in this figure and then overlay this outline in the other plots. This would make these subsequent figures much more useful. Or alternatively lose the lat-lon plots entirely and focus on the area-average means for the different regions shown here and assessed in figure 7.

L115-145 – Is there any validation of this modeling approach? If not this is a major shortcoming. 2 previous papers are cited; if they contain a validation of this approach that should be brought up and explained.

L116-130 – Some sense of the sensitivity of the method to this sampling of glaciers is needed. I did not find the values in the table easy to read so I couldn't tell from looking at this how large the differences were; perhaps a figure would help to illustrate this. Can you be sure that these differences are systematic regional differences rather than the quirks of individual glaciers? Why not take the average over all glaciers rather than just the 3 nearest? These questions ought to be answered here or else citations which address these questions cited. This is another source of uncertainty in the model projections but it is unclear how large it is.

L140-144 – The justification for excluding precipitation here is incomplete (as it is in section 4). What matters is not the fractional precipitation change but rather the ratio of a x dT to b x dP. If you have the beta values it should be straightforward to make a rough calculation of b x dP so that a comparison can be made. This issue is critical to

address properly as your model cannot distinguish GHG from solar forcing (as it uses dT alone) and solar geoengineering has a distinct effect on precipitation.

L146-189 – Is there a validation for this approach, i.e. do the down-scaled, lapse-rate adjusted reanalysis data match observational data?

L148 – "the beginning years" not clear

L150-151 – This sounds like you don't believe them, explain the limits of this approach or rewrite this sentence.

L159 – "and THE INCREASE IN greenhouse gas forcing" rather than the total forcing relative to pre-industrial.

L159-161 – Here and elsewhere need to make it clear that different models had a very different temperature, etc. response to G4's 5Tg.

L162 – No. Injection stops which means the aerosol forcing will decay rather than instantly disappear.

L171 – be more specific, e.g. resolutions of ∼2 degrees.

L176 – Does the glacier not have an altitude of its own recorded somewhere? How large could the difference between the nearest grid-box and the glacier's actual altitude be? I suspect this could be quite large. Has this been dealt with by other papers / methods? Is there any way to get a sense of this?

L191-222 – Why is a regional analysis here? The regional analysis of the glacier response is made much later, it would seem sensible to present this material alongside that.

L191-222 – Some warning should be given about reading too much into the small scale "features" in these plots. These are artifacts of the ensemble average of mismatched, coarse model grids that have been downscaled – no individual model has the kind of spatial variation seen in figure 3a for example.

L193 – estimate = calculate.

Figure 2 and throughout – It is not clear whether the differences plotted in this figure and in all other figures are due solely to differences in the response to the experiments as is implicitly assumed or whether the different make-ups of the ensembles used in each experiment has affected the results substantially. Some measure of this effect should be made.

Figure 2 and other map plots – The continuous color scale is hard to read, is it possible to use a banded scale or to add contour lines?

Figure 2 and other map plots – I'd suggest removing the stippling. First, I'm skepti-cal whether the measure plotted is useful, i.e. does it really give a good sense of the robustness of the regional results? Second, looking at these plots the stippling is ap-plied at a scale much finer than the resolution of the models that generated the results. Third, it's entirely absent from some figures suggesting that the stippling code was not implemented appropriately.

Figure 2 and 3 – Are these annual-mean or JJA-mean results?

L197 and throughout – Are mean +/- standard deviation (which I presume these are) measures appropriate when the ensemble consists of only 4 models? Perhaps mean (min, max) may be more appropriate and informative.

Table 3 – I'd recommend adding much more information here and converting this into a figure which covers the full range of the model responses for each experiment across the different glacier regions. This would help to address my concern about the absence of precipitation from the simulations and the poor justification for why this was left out. Reporting the changes in precipitation in percentage change may help or else this could be combined with an estimate of the beta parameter for glacier sensitivity to precipitation (perhaps expressed as a fraction of the temperature response) to give a quantified estimate of how significant the precipitation response is. Without such

information and given that precipitation is excluded from the glacier model I don't see why devoting a page to the precipitation response was necessary.

Figure 3 – These precipitation results seem to have been down-scaled, how was this done?

L218 – "Precipitation change ratios"? not clear what this means.

L225 – "projections ensembles"?

L233-241 – How does these results compare to previously published results? Perhaps cite the work that made the initial analyses of G3 and G4.

L233 – Should "and the highest rate" read "at the highest rate"?

L238 – 1.7C over what period?

L239-241 – Why does this happen? Here and elsewhere more care should be taken to make clear that G4 produced very different forcing and hence temperature responses in the different models. Some evaluation of how effective G4 was at cooling the climate in different models should be made or reference should be made to a study which makes this analysis.

Figure 4 – There is a suspicious degree of agreement between the models in panel d. These models presumably have quite different TCRs, surely there would be greater differences than this at the regional scale. Please cite some other work or present some evidence that this is not a processing error on your behalf.

Figure 5 – Here and elsewhere the comparison is not simply between different experiments but between different ensembles of models. Some assessment of the significance of this difference ought to be made. A similar line plot like this would be a good place to do so.

Figure 5 – Rather than going into the spatial pattern of the precip response in figure 3 (which is not used) why not add 2 panels on the precip response to this line plot?

Alternatively, do the regional-average analysis I suggested earlier.

L249 – "averaged over the grids" what does this mean?

L263-264 – Explain why this happens. I presume it's because the most sensitive glaciers have already retreated / disappeared but this isn't mentioned anywhere. I'm wondering idly whether an overall sensitivity could be calculated.

L269 – explain why G3 > G4.

L270-280 – There is too much reporting of values and not enough explanation in this paragraph.

L283-284 – Is this area-volume relation surprising? Aren't they explicitly linked in the model? Figure 7 – This regional break-down is great, however it is a shame that none of the previous results are presented on the same basis. This regional (and sub-regional) analysis is much more useful than the spatial maps shown elsewhere, could temperature and precipitation responses be plotted for these regions?

L288-301 – Why isn't the regional climate analysis here with the regional glacier analysis.

L303-359 – It is unclear what this section is for. I'd suggest reframing this as a regular discussion section and broadening its scope to cover all the shortcomings of this study.

L304-306 – This sentence is unclear.

L315-320 – why is this included? Your simulations excluded precipitation effects.

L328-333 – This paragraph is unclear, it is hard to follow the references.

L334-338 – This is also unclear.

L339-342 – Does it? You have not shown the relative significance of precipitation to temperature in this study and table 3 only reports the area-average results which differ from the regional responses.

[Figure]

L339 – if A1B shows a significant trend then surely RCP 8.5 would too given their similarity?

L344-346 – This sentence is unclear and it's not clear which observational data is referred to here.

L350-352 – I'm not sure how useful this measure is given the shortcomings in the approach and the systematic over-estimation due to the exclusion of precipitation. Also, extending the simulations out to 2150 could give rise to the problem that your simple approach to uncertainty bounds would include negative glacier mass.

L361-end – Much of the material presented in this conclusion is new, why does this not appear earlier?

L372 – why is G3 extreme? There are more extreme possibilities. This is an arbitrary scenario.

L383-385 – This is flatly wrong as written here. This result is scenario dependent. If a greater cooling were exerted by solar geoengineering more glaciers would be saved.

---

## Referee Comment (RC2) · Anonymous Referee #2 · 2 Dec 2016

In this study, the authors propose to drive a minimal glacier model with GCM projections in the HMA region. The innovative part of the study is that they assess the impact of geoengineering on glacier changes, which is (as far as I am aware of) not discussed very often. However, the study suffers from the over-simplification of the glacier processes and from poor uncertainty assessments, two points which have to be addressed before considering publication.

[Figure]

**General comments**

*Glacier model*

The glacier model used in this study is quite far behind today's standards (e.g. Marzeion et al., 2012, Huss and Hock, 2015). I list here the major issues that need to be addressed:

- the model only considers changes in ELA with respect to summer temperature. They justify their choice by saying that most glaciers in the region are of the summer accumulation type (which is not proven) and that precipitation varies little over the entire HMA (which is a qualitative statement, and also probably not true for the sub-regions, as shown in Fig. 3). Precipitation has to be considered by the model, and not only summer precipitation: winter precipitation and the differenciation between liquid and solid precipitation has to be taken into account (in particular for the whole western and northern part of the study region, where precipitation is not falling in summer)

- the response time of glaciers has to be taken into account. This has to be parameterised in the volume-area scaling relation, as discussed by Marzeion et al., (2012) and Bahr et al., (2015).

- it is not clear to me how glaciers are supposed to grow in this model. Many glaciers in the HMA are currently growing or at least stagnating (without mentioning debris-covered glaciers), ad point which is not discussed in the study.

- the calibration of the mass-balance (MB) gradients is extremely loose. If I understand well, the MB gradients are defined for one glacier wih observations and then applied to the entire sub-region. By looking at Table 1 (where the MB gradients are described), it looks very unlikely that there is any reason for the local

MG gradients (which contain arbitrary altitude thresholds and other local properties) to be representative for the region. Here I suggest to use either data-driven gradients (i.e. based on climate data) or even much simpler statistical gradients models which would be easier to cross-validate (see validation section below).

*Validation and uncertainty assessment*

The current approach to uncertainty assessment is not robust enough. Validation (i.e. comparison against observations) is quasi non-existent. I agree that given the few number of observations, the task is not trivial. But especially in this case, it is recommended to make full use of all available data:

- the authors could make use of cross-validation to assess the impact of interpolating the gradients on mass-balance (see e.g. Michaelsen, 1987)

- several recent publications made use of satellite observations to assess geodetic MB (e.g. volume changes) in HMA. This could serve as basis for a region-wide validation during the last decade, if only qualitative. See e.g. Huss and Hock (2015) who made use of the region-wide estimates of Gardner et al. (2013)

- the spread between the GCM ensemble members should also be discussed, as it probably impacts the results a lot

**Specific comments**

Add uncertainty ranges to numbers in the abstract

L50: add references to the summer-accumulation type statement (e.g. Fujita, 2008). Besides, it is highly speculative (and probably wrong) to say that all glaciers in HMA

are "mainly" of this type. See the classifications by Rupper and Roe (2008) or the classification by Maussion et al., (2014), which shows that large parts of HMA are not of the summer accumulation type.

L85: I don't understand the need to use different inventories in this study. It seems much more consistent to stick to one, and give all the figures for the one judged more adapted.

L90: please justify your choice of the median for the ELA proxy. What consequences does this choice have in the case of glaciers which are far from equilibrium, as it is the case in Eastern Himalaya?

L99-100: rephrase

Table 1: explain the gradients column in the legend, specify units

L120: reformulate "to calculate two or three SMB gradients with altitude", which is unclear to me

L125: volume area scaling must be extended with a relaxation time scale! See Marzeion et al., (2012) and Bahr et al., (2015).

L127: "by assuming all the decrease in area takes place in the lowest parts of the glacier": but how do you deal with growing glaciers?

L143: "relatively small (<10%).": I wonder as to which percentage the authors would consider that the preciptiation changes aren't "relatively small" anymore. I personnally find that 10% is quite a big deal.

L150: why not considering CRU (https://crudata.uea.ac.uk/cru/data/hrg/), which has a resolution of 0.5deg?

L166: how are they different?

At the end of the methods section the reader is left with many questions about how the

calibration of the $\alpha$ parameter is done, and how the uncertainties are handled in the study.

Fig 2 Fig 3: please make a figure following today's standards. Add country borders or topography (or anything that helps for orientation). Consider using discrete levels instead of continuous colors. Are the anomalies for the entire year or just the summer season?

Fig 5: add the spread between the ensemble members

Fig 6: the uncertainty associated with the various ensemble members should also appear in the spread

L317: deep convection

Conclusions: part of the conclusions should be extended and moved to the discussion (in particular the comparison with other studies).

L368: specify what "close" means

*References*

Bahr, D. B., Pfeffer, W. T. and Kaser, G.: A review of volume-area scaling of glaciers, Rev. Geophys., 95–140, doi:10.1002/2014RG000470, 2015.

Fujita, K. and Ageta, Y.: Effect of summer accumulation on glacier mass balance on the Tibetan Plateau revealed by mass-balance model, J. Glaciol., 46(153), 244–252, doi:10.3189/172756500781832945, 2000.

Gardner, A. S., Moholdt, G., Cogley, J. G., Wouters, B., Arendt, A. a, Wahr, J., Berthier, E., Hock, R., Pfeffer, W. T., Kaser, G., Ligtenberg, S. R. M., Bolch, T., Sharp, M. J., Hagen, J. O., van den Broeke, M. R. and Paul, F.: A Reconciled Estimate of Glacier Contributions to Sea Level Rise: 2003 to 2009, Science., 340(6134), 852–857,

doi:10.1126/science.1234532, 2013.

Marzeion, B., Jarosch, a. H. and Hofer, M.: Past and future sea-level change from the surface mass balance of glaciers, Cryosph., 6(6), 1295–1322, doi:10.5194/tc-6-1295-2012, 2012

Maussion, F., Scherer, D., Mölg, T., Collier, E., Curio, J. and Finkelnburg, R.: Precipitation Seasonality and Variability over the Tibetan Plateau as Resolved by the High Asia Reanalysis*, J. Clim., 27(5), 1910–1927, doi:10.1175/JCLI-D-13-00282.1, 2014.

Michaelsen, J.: Cross-validation in statistical climate forecast models, J. Clim. Appl. Meteorol., 26(11), 1589–1600, doi:10.1175/1520-0450(1987)026<1589:CVISCF>2.0.CO;2, 1987.

Rupper, S. and Roe, G.: Glacier Changes and Regional Climate: A Mass and Energy Balance Approach, J. Clim., 21(20), 5384–5401, doi:10.1175/2008JCLI2219.1, 2008.

---

## Author Comment (AC1) · 9 Mar 2017

In the reply, the referee's comments are in *italics*, our response is in normal text, and quotes from the manuscript are in blue.

***Anonymous Referee #1***

*General comments*
*This paper presents the first analysis of the Himalayan glacier response to solar geoengineering, analyzing the response to a number of scenarios of future greenhouse gas emissions and stratospheric aerosol geoengineering. The authors employ a mixed empirical / statistical model of glacier change driven by temperature change alone and use statistical down-scaling of the GeoMIP multi-model ensemble to produce the input data for their model. They find that solar geoengineering could slow the rate of retreat of Himalayan glaciers and that this benefit would be lost were solar geoengineering to be terminated.*

*This study is a novel contribution to the geoengineering literature and addresses an important issue; however, there are a number of shortcomings in the study that need to be addressed.*

*Most importantly, the method employed by the authors excludes precipitation which is the most critical climate difference between scenarios which include solar geoengineering and those which don't. Studies have reported substantial reductions in monsoon precipitation in scenarios which include solar geoengineering and so it seems odd to exclude this from the analysis. The authors note that excluding precipitation will likely mean that their estimate of the efficacy of solar geoengineering is an overestimate but it is not clear the magnitude of this overestimate; is it of order 1%, 10%, 100%?*

*I'd strongly recommend that the authors either reintroduce precipitation back into their methodology (the model described can use precipitation but it was excluded) even if this is only to provide a robust sensitivity estimate that justifies its exclusion, and I'd suggest doing this even if there are serious shortcomings in the down-scaling method for precipitation or other methodological concerns. Failing that, a much more detailed justification for excluding precipitation should be made along with some robust regional estimates of the significance that the changes in precipitation should be expected to have. The authors note that Rupper and Roe (2008) calculated zonal-mean sensitivities of SMB to temperature and precipitation; I suggest using these to estimate the sensitivity. This study should make clear what the research needs are to make a better estimate of the effects of solar geoengineering on Himalayan glaciers and the significance of including precipitation is a critical question that this paper does not address well.*

**Reply:** In the revision, we include precipitation in our methodology and consider its impact on glacier change. We used zonal-mean sensitivities of SMB to temperature and precipitation calculated by Rupper and Roe (2008). We have done the simulations with precipitation using all models under all the scenarios.

We also add a whole new section on the uncertainties in the projections (section 5) and on the method validation section 3.3.

*I had a number of questions about the validity of the methodology adopted in this study*

*that either require the authors to change the methodology, present some additional justification for their approach, or provide some sense of the sensitivity of their approach to these issues. I list the most significant ones here but there are other minor concerns listed afterwards.*

*First, the authors make comparisons between the responses of a changing ensemble of models for different experiments. It is unclear how much of the difference between these experiment-ensemble means is due to the changing make-up of the ensembles and how much due to robust differences in response. This is particularly concerning in the case of G4 were the models produced very different temperature responses due to their differing treatments of the 5Tg of injected SO2.*

**Reply:** In the revision, we have done all the simulations using every GCM ensemble member and ensemble-mean under every climate scenario. We list the result by every model under all the scenarios in Table 4 in the revision. We also test the significance of the ensemble member differences.

Volume loss using the climate projected by HadGEM2-ES under G4 is far less than that by other models (Table 4), so we exclude it when calculating the G4 model mean.

We tested the differences between RCP8.5, RCP4.5 and G4 using the 4 models in common (Table 4), and find the glacier responses are significantly different (p<0.05). Although there are too few models in common between G3 and G4, the dominant influence of summer melting to the mass balance (Zhao et al., 2016), and the clear difference in temperature across HMA between G3 and G4 (Figs. 2,3) suggest the glacier response in HMA is different between G3 and G4.

*Second, the authors don't do enough to validate their approach. A comparison against historical data or a more explicit reference to studies which validate this approach is needed. Does this model predict stable, growing or shrinking glacier mass for pre-industrial climate conditions? Is this model in agreement with other similar models?*

**Reply:** Our model estimates were already discussed in comparison with estimates for RCP8.5 and 4.5 by Radić et al. (2014) and Marzeion et al. (2012) in the conclusion section of previous manuscript, and we also have them in Section 5.2 of the revision. This is the first simulation of geoengineering forcing of HMA glaciers.

In our method, we need to use glacier outline and glacier surface elevation to estimate glacier mass balance. Because we do not have such input data for the pre-industrial period, we cannot do simulations of glacier change for pre-industrial climate conditions.

In the revision, we added a new Section 3.3 which is about validation of the glacier model.

**3.3 Validation of the glacier model and methodology**

[revised manuscript text omitted]

*Third, the authors need to fix parameters for the thousands of glaciers in their study area but only have 13 glaciers to base these parameters on. As they are making a regional analysis they use only the 3 nearest glaciers to fix parameters in a regional manner. It is not immediately clear how different these 13 glaciers are from the results presented and so it is not clear how sensitive the results are to this sampling. The authors don't discuss whether one could expect the sampled glaciers to be representative of their region or why one would expect robust regional differences between the glaciers' behavior.*

**Reply:** Although the number of sampled glaciers is only 13 and the glaciers are randomly located, we find interestingly that the SMB gradients of the few glaciers in one sub-region are similar in their common elevation range. Again in Section 3.3:

For inner Tibet, there are 3 glaciers (Zhadang, Gurenhekou and Xiao Dongkemadi Glacier) with SMB observations, and they have almost the same SMB-altitude gradients, 0.0041 m m$^{-1}$, over their common elevation range (5515~5750 m, Table 1); two glaciers (Naimona'nyi and Kangwure) in central Himalaya have SMB gradients of 0.0038 m m$^{-1}$ in their common altitude range of 5700~6100 m. These similarities suggest that the measured glaciers share some important characteristics with the vast majority which are not surveyed.

In the revision, we improve Table 1 to make this clearer.

*Finally, the authors downscale temperature to a high-resolution grid however they don't make any assessment of whether this downscaled temperature data matches observations. I'd expect that there could be quite large differences between the real glacier altitude and the altitude of the nearest high-resolution grid point given the complex topography. I was also wondering given that the observed altitude of the glaciers is used to calculate the ELA, why not use it directly to perform this lapse-rate adjustment?*

**Reply:** We guess the referee did not get what we did in the data-downscaling. In fact, we did the lapse-rate adjustment for temperature data just as the referee suggests. In the revision, we used temperature data from CRU and precipitation data from GPCC, which

have good reputation of the data quality. We have done many changes in Section 3.2 and 3.3 on downscaling that hopefully help clarify it:

The temperature and precipitation on each glacier were calculated by an altitude temperature lapse rate of 0.65℃/100 m, precipitation lapse rate of 3%/100 m, and the elevation difference of the glacier surface elevation relative to the nearest fine grid point.

In our simulations we have used constant lapse rates for temperature (0.65℃/100 m) and precipitation (3%/100m). To check how reliable this is we chose 5 meteorological stations close to glaciers and calculated correlation coefficients for JJA temperature and annual precipitation at the station and at the nearest downscaled grid point from 1980 to 2013 (n=34). Precipitation correlations were higher than 0.85 for all the stations (p<0.001), while temperatures correlations were 0.47-0.85 (p<0.01).

*Relatedly, I'd recommend that the authors broaden the uncertainties section into a proper discussion that covers the shortcomings of the methodology employed and discusses the sensitivity of the findings to various assumptions. A key question which should motivate the material in this section is – how would the glacier response in a model with a full surface mass balance treatment driven directly by high-resolution climate data differ?*

**Reply:** We do broaden the uncertainties section (see Section 5 in the revision) to discuss the uncertainties caused by climate forcing and by glacier model. In the revision, we use the relatively high resolution, monthly-mean gridded 0.5°×0.5° temperature data from the CRU TS 3.24 dataset instead of the 1°×1° temperature data from Berkeley Earth project, which was used before. We also discuss possible dynamic downscaling for the glacier model in section 5.1:

The models are also relatively coarsely gridded, certainly compared with the vast majority of glaciers, and so differences may be expected between statistically downscaled forcings based on lapse rates that we use here and that produced from high resolution dynamic climate model forcing.

we note that the distribution of meteorological stations in the study region is very sparse, especially in the northwest of this region (Liu and Chen, 2000). Therefore, both the CRU gridded data and data from models projections that we used in this study may have low accuracy for specific glacier regions. This has also implications for the use of very high resolution dynamic models; one such model simulated air temperatures and down-welling radiative fluxes well, but not wind speed and precipitation, producing unstable results when used with the CLM45 land model that simulated ground temperatures and snow cover (Luo et al., 2013). Explicit glacier atmospheric mass balance modelling (Mölg et al., 2013), a technique based on very high spatial and temporal resolution climate data (hourly and 60 m) was used on Zhadang glacier (Fig. 1, Table 1) with in-situ observations available, but not across the general expanse of the glaciated region; this study also noted the importance of wind speed to glacier mass balance in the region influenced by the Indian monsoon. Maussion et al., (2013) demonstrate that 10 km resolution dynamic modelling of the region can be done

successfully, and potentially can improve the precipitation modelling over the statistical downscaling methodology we employ here, though to date this is a reanalysis dataset with no prognostic simulations. Zhao et al. (2014 and 2016) used a 25 km resolution regional climate model RegCM3 to drive their simulations of glacier response to scenario A1B. By 2050 under A1B (which is intermediate between RCP4.5 and 8.5 in temperature rises), a sea level rise equivalent to 9.2 mm was projected from HMA. In comparison, our estimates are 11.1-12.5 mm for RC4.5 and 8.5 (Fig. 4).

*I had a few concerns about the structure and the focus of the paper that relate to the regional climate analysis. First, it seems strange to separate the regional climate analysis from the regional glacier analysis. Why not start with the whole-region analysis of climate and glacier response and then conduct a regional climate and glacier analysis afterwards? Second, Figures 2 and 3 seem of very little use as there are no cues on these plots to relate the results presented to the regional glacier analysis in figure 7. Given glacier response is calculated across the region or over sub-regions, I'd recommend doing a sub-regional-mean analysis as in figure 7 instead of including figures 2 and 3. This would also help to make clearer how significant excluding precipitation from the analysis was.*

**Reply**:Agreed.

Firstly, we change the structure of the result section (Section 4) as follows:

4 Result

4.1 Climate and glacier change across HMA

    4.1.1 Temperature and precipitation over HMA

    4.1.2 Glacier change across HMA

4.2 Sub-regional climate and glacier changes

    4.2.1 Sub-regional temperature and precipitation change

    4.2.2 Sub-regional glacier changes

Secondly, we changed Fig. 2 and 3 to sub-regional analysis of temperature and precipitation (Fig. 3 in the revision), which are similar and have better connection to Fig. 7 --- the sub-regional analysis of glacier volume change.

*Finally, the conclusion needs to be revised as it presents a lot of new results and material, some of which are not appropriate for a conclusion.*

**Reply:** We rewrote the whole conclusion section 6, and moved some material to the Uncertainty section 5.

***Specific comments***

*L18-21 – unclear sentence: 5 under G4 what?*

**Reply:** Changed to five models under G4 and six models under RCP4.5 and RCP8.5.

*L23-25 – I don't recall this point being made in the paper.*

**Reply:** We removed this point in the revision.

*L30-31 – This is a far-reaching general statement extrapolated from a specific scenario.*

*It holds for the specific case but not for the general case.*
**Reply:** We remove this sentence.

*L55 – delete hence*
**Reply:** Done.

*L55-57 – These are very different types of reasons and I don't see how the second matters.*
**Reply:** We rewrite this sentence as "the glaciers are affected by both the South Asian monsoon system and the westerly cyclonic systems, depending on specific location across the region, thus the region integrates the climate response to two important global circulation systems (Mölg et al., 2013)."

*L57-60 – Irvine et al. 2012 is a semi-empirical study and McCusker et al. discussed ice-sheet implications of their climate model results but did not simulate ice-sheet or sea-level response at all.*
**Reply:** We move Irvine et al. (2012) to the correct place. We change this sentence to "glacier responses to geoengineering scenarios has been limited to studies on global responses based on semi-empirical models (Moore et al., 2010;Irvine et al. 2012) or from simplified ice sheet responses (Irvine et al. 2009; Applegate et al., 2015) or implications of climate model (McCusker et al., 2015), with nothing to date on mountain glacier impacts."

*L75 – "regions of newly defined regions of " – fix*
**Reply:** modified to "defined regions of".

*L86-89 – This sentence is unclear. Why is this simple? What issues could this pose?*
**Reply:** All the glacier outlines data are takes from RGI 5.0 dataset. The Second Chinese Glacier Inventory and the "Glacier Area Mapping for Discharge from the Asian Mountains" (GAMDAM) inventory provide data in certain regions in RGI 5.0. As the referee asked below, we improve this paragraph to make this meaning clear.
Notice the date of the data are different in different regions. And the date range for each region is short – a few years. So we take the same date for the glaciers in one region from the same data source. We do not think it would bring big issue. We add "Because the data range from one data source is only a few years" here in the revision.
The Randolph Glacier Inventory (RGI) database contains outlines of almost all glaciers and ice caps outside the two ice sheets (Arendt et al., 2015). Our study region covers HMA (26–46° N, 65–105° E), which corresponds to the defined regions of Central Asia, South Asia West and South Asia East in the RGI 5.0. According to the RGI 5.0, the study region contains a total of 94,000 glaciers and a glaciered area of about 110,000 km2. The RGI 5.0 data inside China are based on the Second Chinese Glacier Inventory (Guo et al., 2015), which provides glacier outlines from 2006–2010, except for some older outlines from the First Chinese Glacier Inventory where suitable imagery could not be found - mainly in southern and eastern Tibet (the S and E Tibet RGI 5.0 subregion), most of which were made in the 1970s. The RGI 5.0 data outside China are from the "Glacier Area Mapping for Discharge from the Asian Mountains" (GAMDAM) inventory (Nuimura et al., 2015) and nearly all come from 1999–2003 with images selected as close to the year 2000 as possible. Because the data range from each data source is only a few years, we take three reference years: 1980, 2009, and 2000, as start dates for our model simulations of glaciers in S and E Tibet, elsewhere in China, and outside China, respectively.

*L90-95 – This approach needs some justification and citations – why is it appropriate? Is this a widely accepted approach? Etc.*
**Reply:** Using median altitude as a proxy of ELA is a widely accepted approach (e.g. Nuimura et al., 2015).
We add this in the revision:
Following previous authors (Nuimura, 2015; Zhao et al., 2016), we use median altitude from RGI 5.0 for each glacier as a proxy for equilibrium line altitude (ELA).
The sensitivity is discussed in section 3.3
In choosing the initial ELAs for each glacier, there are several reasonable alternatives (Zhao et al., 2016): i) using ELAs interpolated from the first Chinese glacier inventory, ii) median elevations from RGI dataset, iii) the elevation of the 60th percentile of the cumulative area above the glacier terminus. These three choices lead to a range of about 2.5 mm of global sea level in glacier volume loss at 2050. In this study, we use median elevations from RGI dataset, which corresponds to the median result.

*L98-99 – What precisely is done? I believe this is explained in the next section, if so make that link explicit.*
**Reply:** The parameterizations of mass balance with altitude relative to the ELA is precisely written in Zhao et al. (2014). We add this reference here.

*Fig 1 + other lat-lon plots – It would help to make sense of the lat-lon plots if there was some frame of reference beyond the lat-lon coordinates. Would it be possible to mark the numbered regions (referred to later) with boxes or outlines in this figure and then over lay this outline in the other plots. This would make these subsequent figures much more useful. Or alter natively lose the lat-lon plots entirely and focus on the area-average means for the different regions shown here and assessed in figure 7.*
**Reply:** We focus on the area-average means for the different regions. As the referee suggested here and afterward, we change Fig. 2 and 3 (the spatial maps of temperature and precipitation) to sub-regional analysis of temperature and precipitation.

*L115-145 – Is there any validation of this modeling approach? If not this is a major shortcoming. 2 previous papers are cited; if they contain a validation of this approach that should be brought up and explained.*
**Reply:** Section 3.3 Validation of the glacier model. And Section 5.2 on uncertainties discuss these issues in detail and at length.

*L116-130 – Some sense of the sensitivity of the method to this sampling of glaciers is needed. I did not find the values in the table easy to read so I couldn't tell from looking at this how large the differences were; perhaps a figure would help to illustrate this. Can you be sure that these differences are systematic regional differences rather than the quirks of individual glaciers? Why not take the average over all glaciers rather than just the 3 nearest? These questions ought to be answered here or else citations which address these questions cited. This is another source of uncertainty in the model projections but it is unclear how large it is.*

**Reply:** We do not quite understand all the points in this question. Hopefully the changes in Section 3 make the method clearer, along with the cited reference to Zhao et al 2014. We tried to clarify our method, in section 3.3, in relation to mass balance gradients we show. The point on sensitivity to SMB, Zhao et al 2016, showed that it is important, but where the data allow the gradients to be checked within a sub-region, they agree very well between glaciers:

With so few glacier observations available, there is an issue of how representative they are of the general population. For inner Tibet, there are 3 glaciers (Zhadang, Gurenhekou and Xiao Dongkemadi Glacier) with SMB observations, and they have almost the same SMB-altitude gradients, 0.0041 m m-1, over their common elevation range (5515~5750 m, Table 1); two glaciers (Naimona'nyi and Kangwure) in central Himalaya have SMB gradients of 0.0038 m m-1 in their common altitude range of 5700~6100 m. These similarities suggest that the measured glaciers share some important characteristics with the vast majority which are not surveyed.

*L140-144 – The justification for excluding precipitation here is incomplete (as it is in section 4). What matters is not the fractional precipitation change but rather the ratio of a x dT to b x dP. If you have the beta values it should be straightforward to make a rough calculation of b x dP so that a comparison can be made. This issue is critical to address properly as your model cannot distinguish GHG from solar forcing (as it uses dT alone) and solar geoengineering has a distinct effect on precipitation.*

**Reply:** Agreed.

In the revision, we include precipitation.

*L146-189 – Is there a validation for this approach, i.e. do the down-scaled, lapse-rate adjusted reanalysis data match observational data?*

**Reply:** Yes. To validate the downscaled, lapse-rate adjusted reanalysis data, we choose 5 climate stations close to glaciers and calculated correlation coefficients for the station JJA temperature and annual precipitation and the nearest downscaled gridpoint from 1980 to 2013 (n=34). Precipitation correlation was higher than 0.85 for all the stations (p<0.001), while temperatures correlations were 0.47-0.85 (p<0.01).

In the revision, we now use the relatively high resolution, monthly-mean gridded 0.5° ×0.5° temperature data from CRU TS 3.24 dataset (Harris et al., 2014), instead of 1° ×1° temperature data from Berkeley Earth Project, because CRU has better reputation

of data quality control. We use 0.5°×0.5° monthly total gridded precipitation data from the Global Precipitation Climatology Centre (GPCC) Total Full V7 dataset (Becker et al., 2013).

*L148 – "the beginning years" not clear*
**Reply:** We run the simulations for glacier change from the relevant start years (Section 2) to the year 2089.

*L150-151 – This sounds like you don't believe them, explain the limits of this approach or rewrite this sentence.*
**Reply:** In the revision, we use temperature data from CRU instead of Berkeley Earth Project. So we rewrite this sentence as "From the start years to 2013, we use the relatively high resolution, monthly-mean gridded 0.5°×0.5° temperature data from the CRU TS 3.24 dataset (Harris et al., 2014)".

*L159 – "and THE INCREASE IN greenhouse gas forcing" rather than the total forcing relative to pre-industrial.*
**Reply:** Done.

*L159-161 – Here and elsewhere need to make it clear that different models had a very different temperature, etc. response to G4's 5Tg.*
**Reply:** OK. We add this in the revision "The across model spread of temperatures under G4 is larger than under e.g. RCP4.5, (there are too few ensemble member models under G3 to see this) because of differences in how the aerosol forcing is handled, and each model has a different temperature response to the combined long and shortwave forcing (Yu et al., 2015)."

*L162 – No. Injection stops which means the aerosol forcing will decay rather than instantly disappear.*
**Reply:** OK. We delete "driven by forcing from RCP4.5 alone".

*L171 – be more specific, e.g. resolutions of ~ 2 degrees.*
**Reply:** We show the resolution in Table 2. We add "(Table 2)" here.

*L176 – Does the glacier not have an altitude of its own recorded somewhere? How large could the difference between the nearest grid-box and the glacier's actual altitude be? I suspect this could be quite large. Has this been dealt with by other papers /methods? Is there any way to get a sense of this?*
**Reply:** We rewrite this sentence to make it clear. The individual glacier has its own altitude. There is difference between the nearest grid-box and the glacier's actual altitude. The temperature and precipitation on each glacier were calculated by an altitude temperature lapse rate of 0.65℃/100 m, precipitation lapse rate of 3%/100m, and the elevation difference of the glacier surface elevation relative to the nearest fine gridpoint.

*L191-222 – Why is a regional analysis here? The regional analysis of the glacier response is made much later, it would seem sensible to present this material alongside that.*

**Reply:** We changed the structure of the paper in the revision.

*L191-222 – Some warning should be given about reading too much into the small scale "features" in these plots. These are artifacts of the ensemble average of mismatched, coarse model grids that have been downscaled – no individual model has the kind of spatial variation seen in figure 3a for example*

**Reply:** OK. But as the referee suggested, we changed Fig. 3 to sub-regional analysis.

*L193 – estimate = calculate.*

**Reply:** We change "estimate" to "calculate".

*Figure 2 and throughout – It is not clear whether the differences plotted in this figure and in all other figures are due solely to differences in the response to the experiments as is implicitly assumed or whether the different make-ups of the ensembles used in each experiment has affected the results substantially. Some measure of this effect should be made.*

**Reply:** As the referee suggested, we changed Fig. 2 to sub-regional analysis (Fig. 3 in the revision).

There are 4 models in common under RCP8.5, RCP4.5 and G4. We tested the differences between RCP8.5, RCP4.5 and G4 using the 4 models in common (Table 4), and find the glacier responses are significantly different (p<0.05). There are only 2 common ensemble members (BNU-ESM, HadGEM2-ES) used in both G3 and G4, but we exclude HadGEM2-ES because this model gives a very different result from other models under G4. So there is only BNU-ESM in common under G3 and G4.

*Figure 2 and other map plots – The continuous color scale is hard to read, is it possible to use a banded scale or to add contour lines?*

**Reply:** As the referee suggested, we changed Fig. 2 from a map to a sub-regional line plot (Fig. 3 in the revision).

*Figure 2 and other map plots – I'd suggest removing the stippling. First, I'm skeptical whether the measure plotted is useful, i.e. does it really give a good sense of the robustness of the regional results? Second, looking at these plots the stippling is applied at a scale much finer than the resolution of the models that generated the results. Third, it's entirely absent from some figures suggesting that the stippling code was not implemented appropriately.*

**Reply:** As the referee suggested, we changed Fig. 2 and Fig. 3 to sub-regional line plots (Fig. 3 in the revision).

*Figure 2 and 3 – Are these annual-mean or JJA-mean results?*

**Reply:** They are JJA-mean temperature in Figure 2 and annual precipitation in Figure 3.

*L197 and throughout – Are mean +/- standard deviation (which I presume these are) measures appropriate when the ensemble consists of only 4 models? Perhaps mean (min, max) may be more appropriate and informative.*
**Reply:** For simplicity we prefer to stick with standard deviations for the results, and it also means that we can remove outliers such as the G4 result of HadGEM2-ES. We add a new table (Table 4) in the revision to show all model results, and their mean +/- standard deviation under all the scenarios.

*Table 3 – I'd recommend adding much more information here and converting this into a figure which covers the full range of the model responses for each experiment across the different glacier regions. This would help to address my concern about the absence of precipitation from the simulations and the poor justification for why this was left out. Reporting the changes in precipitation in percentage change may help or else this could be combined with an estimate of the beta parameter for glacier sensitivity to precipitation (perhaps expressed as a fraction of the temperature response) to give a quantified estimate of how significant the precipitation response is. Without such information and given that precipitation is excluded from the glacier model I don't see why devoting a page to the precipitation response was necessary.*
**Reply:** In the revision, we include precipitation in the model.

*Figure 3 – These precipitation results seem to have been down-scaled, how was this done?*
**Reply:** we downscale both the CRU gridded temperature data, the GPCC gridded precipitation data and the climate model output to a grid based on a land surface topography having resolution of $0.1126^{\circ} \times 0.1126^{\circ}$ using an altitude temperature lapse rate of $0.65°C/100$ m, an altitude precipitation lapse rate of 3%/100m, and elevation difference of the fine grid relative to the climate model grid.

*L218 – "Precipitation change ratios"? not clear what this means.*
**Reply:** We no longer use this approach in the manuscript.

*L225 – "projections ensembles"?*
**Reply:** We change it to "projections".

*L233-241 – How does these results compare to previously published results? Perhaps cite the work that made the initial analyses of G3 and G4.*
**Reply:** We cite a paper Yu et al. (2015) here, and change this sentence to "There are relative coolings of $1.05°C$ under G3 and $0.76°C$ under G4 compared with RCP4.5 during 2020-2069 across the whole region (Fig. 3). Yu et al. (2015) noted that G3 produced a relative cooling of $0.58°C$ and G4 of $0.53°C$ in globally averaged temperature over the 2030-2069 period."

*L233 – Should "and the highest rate" read "at the highest rate"?*
**Reply:** Yes. We change it.

*L238 – 1.7C over what period?*
**Reply:** The temperature rise is over the period 2070-2089 relative to the period 2050-2069. We add the period in the revision.

*L239-241 – Why does this happen? Here and elsewhere more care should be taken to make clear that G4 produced very different forcing and hence temperature responses in the different models. Some evaluation of how effective G4 was at cooling the climate in different models should be made or reference should be made to a study which makes this analysis.*
**Reply:** We add the reason in the revision. This is due to G4 having a constant stratospheric aerosol injection rate of 5 Tg SO2 per year, while G3 gradual ramps-up the aerosol so that about twice as much is needed by 2069, depending upon the sensitivity of the particular model to stratospheric sulphate aerosols. Hence, the radiative impact of terminating G3 is about twice as large as terminating G4, and the termination temperature signal is much more obvious in G3 than G4.

We made it clear that "The across model spread of temperatures under G4 is larger than under e.g. RCP4.5, (there are too few ensemble member models under G3 to see this) because of differences in how the aerosol forcing is handled, and each model has a different temperature response to the combined long and shortwave forcing (Yu et al., 2015)."

In Yu et al. (2015), they found "Over the period from 2030 to 2069, the global average SAT under rcp45 increased by 0.81 ± 0.21 °C compared with the baseline (average over 2010–2029 under rcp45); while G4, the 40 year annual global mean SAT increased by 0.28 ± 0.31 °C". So the cooling effect is about 0.54°C. Therefore, we add this paragraph in the revision

"There are relative coolings of 1.05° C under G3 and 0.76° C under G4 compared with RCP4.5 during 2020-2069 across the whole region (Fig. 3). Yu et al. (2015) noted that G3 produced a relative cooling of 0.58° C and G4 of 0.53° C in globally averaged temperature over the 2030-2069 period."

*Figure 4 – There is a suspicious degree of agreement between the models in panel d. These models presumably have quite different Transient climate response (TCR)s, surely there would be greater differences than this at the regional scale. Please cite some other work or present some evidence that this is not a processing error on your behalf.*
Reply: The plots in Fig. 4 (which is Fig.2 in the revision) show the average temperatures after grid-point by grid-point bias corrections. Ranges found are slightly smaller than the regional spread found by Yu et al. (2015) due to grid-point by grid-point bias correction we apply here. Under RCP8.5, temperature rises in HMA about 5 °C from the year 2020 to 2089 (panel d), which is higher than the global temperature rise of 3°C

over the same period (IPCC AR5). We do not think we made any error.

*Figure 5 – Here and elsewhere the comparison is not simply between different experiments but between different ensembles of models. Some assessment of the significance of this difference ought to be made. A similar line plot like this would be a good place to do so.*
**Reply:** We add the spread between ensemble members in this plot.

*Figure 5 – Rather than going into the spatial pattern of the precip response in figure 3 (which is not used) why not add 2 panels on the precip response to this line plot? Alter natively, do the regional-average analysis I suggested earlier.*
**Reply:** We add 2 panels on the precipitation response in the revision.

*L249 – "averaged over the grids" what does this mean?*
**Reply:** it means the averaged over the downscaled grid.

*L263-264 – Explain why this happens. I presume it's because the most sensitive glaciers have already retreated / disappeared but this isn't mentioned anywhere. I'm wondering idly whether an overall sensitivity could be calculated.*
**Reply:** Yes: The RCP4.5 and RCP8.5 scenarios produce similar continuous mass loss until approximately 2035 (Fig. 4a) mainly due to the similarity of temperatures projected by RCP4.5 and RCP8.5 in the period 2020-2035 (Fig. 2), and both show relatively slower loss rates after about the year 2050 probably because the most sensitive glaciers have already retreated before 2050.

*L269 – explain why G3 > G4.*
**Reply:** G3 reduces glacier loss more than G4, which is due to stronger temperature cooling effect under G3 (section 4.1.1). We add this in the revision.

*L270-280 – There is too much reporting of values and not enough explanation in this paragraph.*
**Reply:** We add Table 4 to show the result of all the models, and remove some values here.

*L283-284 – Is this area-volume relation surprising? Aren't they explicitly linked in the model?*
**Reply:** It is not surprising. We add "as may be expected" and delete "quite".

*Figure 7 – This regional break-down is great, however it is a shame that none of the previous results are presented on the same basis. This regional (and subregional) analysis is much more useful than the spatial maps shown elsewhere, could temperature and precipitation responses be plotted for these regions?*
**Reply:** In the revision, we add the temperature and precipitation responses in each subregion.

*L288-301 – Why isn't the regional climate analysis here with the regional glacier analysis.*
**Reply:** We have changed the structure of the result section.

*L303-359 – It is unclear what this section is for. I'd suggest reframing this as a regular discussion section and broadening its scope to cover all the shortcomings of this study.*
**Reply:** we did so in the revision.

*L304-306 – This sentence is unclear.*
**Reply:** We improve this sentence as "Firstly, only 3 ESMs participated in G3 but 5 in G4 simply because doing the G3 experiment is difficult and time-consuming to set-up."

*L315-320 – why is this included? Your simulations excluded precipitation effects.*
**Reply:** We include precipitation in the revision.

*L328-333 – This paragraph is unclear, it is hard to follow the references.*
**Reply:** We rephrase this, and split the paragraph into 4 different ones in Section 3.3. The relevant 4 lines are now: In choosing the initial ELAs for each glacier, there are several reasonable alternatives (Zhao et al., 2016): i) using ELAs interpolated from the first Chinese glacier inventory, ii) median elevations from RGI dataset, iii) the elevation of the 60th percentile of the cumulative area above the glacier terminus. These three choices lead to a range of about 2.5 mm of global sea level in glacier volume loss at 2050. In this study, we use median elevations from RGI dataset, which corresponds to the median result.

*L334-338 – This is also unclear.*
**Reply:** We delete this since we include precipitation in the revision.

*L339-342 – Does it? You have not shown the relative significance of precipitation to temperature in this study and table 3 only reports the area-average results which differ from the regional responses.*
**Reply:** We remove this sentence since we include precipitation in the revision.

*L339 – if A1B shows a significant trend then surely RCP 8.5 would too given their similarity?*
**Reply:** We remove this sentence since we include precipitation in the revision.

*L344-346 – This sentence is unclear and it's not clear which observational data is referred to here.*
**Reply:** changed to : The set of parameters we use in this study corresponds to the lower bound of estimated volume loss, but one that is best matched to the observational dataset of 230 separate glaciers (Moore et al., 2013).

*L350-352 – I'm not sure how useful this measure is given the shortcomings in the approach and the systematic over-estimation due to the exclusion of precipitation. Also, extending the simulations out to 2150 could give rise to the problem that your simple approach to uncertainty bounds would include negative glacier mass.*
**Reply:** we removed this measure in the revision.

*L361-end – Much of the material presented in this conclusion is new, why does this not appear earlier?*
**Reply:** we moved them to the uncertainty section (Section 5) before conclusion.

*L372 – why is G3 extreme? There are more extreme possibilities. This is an arbitrary scenario.*
**Reply:** we remove "extreme".

*L383-385 – This is flatly wrong as written here. This result is scenario dependent. If a greater cooling were exerted by solar geoengineering more glaciers would be saved.*
**Reply:** Perhaps we were unclear, we changed the final paragraph to:
Although G3 keeps the average temperature from increasing in the geoengineering period, G3 only slows glacier shrinkage by about 50% relative to losses from RCP8.5. Approximately 72% of glaciated area remains at 2069 under G3 compared with about 30% for RCP8.5. The reason for the G3 losses is likely to be that the glaciers in HMA are not in equilibrium with present day climate, so simply stabilizing temperatures at early 21st century levels does not preserve them. To do that would require significant cooling, perhaps back to early 20th century levels. Achieving that coling by sulphate aerosol injection may not be possible. The 5 Tg of SO2 per year specified in G4 is about the same loading as a 1991 Mount Pinatubo volcanic eruption every 4 yr (Bluth et al., 1992). G3 requires increasing rates of injection, to 9.8 Tg for the BNU-ESM at 2069. As aerosol loading increases, its efficacy decreases as particles coalesce and fall out of the stratosphere faster, while also becoming radiatively less effective (Niemeier and Timmreck, 2015). This effect is so strong that it appears unfeasible to use sulphate aerosols to completely eliminate warming from scenarios such as RCP8.5. Greenhouse gas emissions would require very drastic reduction from present levels, and net negative emissions within the next few decades, to limit global temperature rises to 1.5 or 2°C (Rogelj, et al., 2015). If such targets were met, then it is conceivable that plausible quantities of sulphate aerosol geoengineering may be able to maintain 2020 temperatures throughout the 21st century. Even if this politically very difficult combination of drastic emission cuts and quite aggressive sulphate aerosol geoengineering were done, then our simulations suggest the disappearance of about 1/3 of the glaciated area in HMA by 2069 still cannot be avoided.

---

## Author Comment (AC2) · 9 Mar 2017

In the reply, the referee's comments are in *italics*, our response is in normal text, and quotes from the manuscript are in blue.

***Anonymous Referee #2***

*In this study, the authors propose to drive a minimal glacier model with GCM projections in the HMA region. The innovative part of the study is that they assess the impact of geoengineering on glacier changes, which is (as far as I am aware of) not discussed very often. However, the study suffers from the over-simplification of the glacier processes and from poor uncertainty assessments, two points which have to be addressed before considering publication.*

***General comments***
*Glacier model*
*The glacier model used in this study is quite far behind today's standards (e.g.Marzeion et al., 2012, Huss and Hock, 2015). I list here the major issues that need to be addressed:*
*• the model only considers changes in ELA with respect to summer temperature. They justify their choice by saying that most glaciers in the region are of the summer accumulation type (which is not proven) and that precipitation varies little over the entire HMA (which is a qualitative statement, and also probably not true for the sub-regions, as shown in Fig. 3). Precipitation has to be considered by the model, and not only summer precipitation: winter precipitation and the differenciation between liquid and solid precipitation has to be taken into account (in particular for the whole western and northern part of the study region, where precipitation is not falling in summer)*
**Reply:** Yes the model is relatively simple, but we note that data are limited in HMA so providing verification and calibration for more sophisticated models is problematic (See the new Section 5).
Specifically addressing the issues raised:
We considered the annual precipitation, and the differentiation between liquid and solid precipitation in the revision.

*• the response time of glaciers has to be taken into account. This has to be parameterised in the volume-area scaling relation, as discussed by Marzeion et al., (2012) and Bahr et al., (2015).*
**Reply:** We take into account the response time of glaciers. We add response time in the volume-area scaling relation as in Marzeion et al., (2012) and present it in section 3.1.

*• it is not clear to me how glaciers are supposed to grow in this model. Many glaciers in the HMA are currently growing or at least stagnating (without mentioning debris-covered glaciers), ad point which is not discussed in the study.*
**Reply:** We add in the method section 3.1 a description of how we deal with glaciers growing. Integrating the SMB over each glacier gives the mass balance, which is also the volume change rate, which is converted to an area change rate using volume–area scaling.

The set of glacier surface grid points is updated every year --- the number of the grid points that need to be removed or added is calculated using the area change rate while the elevation of the grid points is updated using SMB.

For advancing glaciers, we add grid points to the glacier surface grid, whose elevations are all supposed to be the glacier elevation minimum in the $n+1_{th}$ year, $z_{min}(n+1)$, which is obtained as follows by assuming a constant glacier surface slope,

$$z_{min}(n+1)=z_{max}(n+1)+\frac{L(n+1)}{L(n)}\bullet(z_{min}(n)-z_{max}(n)),\qquad(5)$$

where $z_{max}(n+1)$ denotes the glacier elevation maximum in the $n+1_{th}$ year. We also limited the maximal surface increase at any point on the glacier to 15 m above the initial elevation at the starting year. We chose to do this because the valley glacier is physically constrained from growing above the level of the surrounding mountain ridge and side-walls.

• *the calibration of the mass-balance (MB) gradients is extremely loose. If I understand well, the MB gradients are defined for one glacier with observations and then applied to the entire sub-region. By looking at Table 1 (where the MB gradients are described), it looks very unlikely that there is any reason for the local MG gradients (which contain arbitrary altitude thresholds and other local properties) to be representative for the region. Here I suggest to use either data-driven gradients (i.e. based on climate data) or even much simpler statistical gradients models which would be easier to cross-validate (see validation section below).*
**Reply:** The SMB gradients are data-based and come from the sparse dataset available, as described in Zhao et al. (2014), with some additional glaciers in this study. We add information about ELA and altitude ranges for each glacier in Table 1. We have only one glacier with SMB measurements in most sub-regions, so we cannot do cross-validate everywhere. However, interestingly, in a few sub-region where there are two or three glaciers, we found that the SMB gradients of these glaciers is very similar in their common altitude range. For inner Tibet, there are 3 glaciers (Zhadang, Gurenhekou and Xiao Dongkemadi Glacier) with SMB observations, and they have almost the same SMB-altitude gradients, 0.0041 m m$^{-1}$, over their common elevation range (5515~5750 m, Table 1); two glaciers (Naimona'nyi and Kangwure) in central Himalaya have SMB gradients of 0.0038 m m$^{-1}$ in their common altitude range of 5700~6100 m. These similarities suggest that the measured glaciers share some important characteristics with the vast majority which are not surveyed.

*Validation and uncertainty assessment*
*The current approach to uncertainty assessment is not robust enough. Validation (i.e. comparison against observations) is quasi non-existent. I agree that given the few*

*number of observations, the task is not trivial. But especially in this case, it is recommended to make full use of all available data:*

*• the authors could make use of cross-validation to assess the impact of interpolating the gradients on mass-balance (see e.g. Michaelsen, 1987)*

**Reply:** We add a section 3.3 that discusses validation for the glacier model. We show that the model produces significant correlations on decadal scales with observations, and also how the benchmark glaciers agree well on MB gradients where they can be compared. We also show in Table 2 how the elevation changes simulated compare with satellite altimetry estimates at a marginally significant level, but which is of course limited in accuracy by the few regions and gross averaging from the satellite data. Section 5 also discusses in depth how climate forcing and the glacier model affect the simulations.

*• several recent publications made use of satellite observations to assess geodetic MB (e.g. volume changes) in HMA. This could serve as basis for a region-wide validation during the last decade, if only qualitative. See e.g. Huss and Hock (2015) who made use of the region-wide estimates of Gardner et al. (2013)*

**Reply:** We have a section about validation of glacier model (section 3.3) in the revision. We also estimated elevation changes for individual glaciers directly from simulated volume and area changes, then calculated the average rate of elevation change for all the glaciers in each sub-region and compared them with remote-sensing estimates from 2003 to 2009 from Gardner and others (2013), Table 2. The correlation coefficient between the Gardner et al. (2013) estimates for the 6 RGI 5.0 sub-regions with data regional and our modeled regional averages is 0.7 which is marginally significant, (p<0.1).

Also note ELA evolution is a key parameter in the method. As a validation of the method, Zhao et al. (2016) calculated the ELA for nine glaciers in China, India and Kyrgyzstan, and compared them with the observed ELA time series by similarities of decadal trends and also annual variability. The ELA parameterization produced reasonable fits to observed ELA decadal trends on 9 glaciers, with a correlation coefficient of 0.6 which is significant (p<0.05, the values we give for p are single tailed Pearson correlation tests).

*• the spread between the GCM ensemble members should also be discussed, as it probably impacts the results a lot.*

**Reply:** We add the simulation results using GCM ensemble members, and discuss the spread between them in the revision.

***Specific comments***
*Add uncertainty ranges to numbers in the abstract*
**Reply:** done.

*L50: add references to the summer-accumulation type statement (e.g. Fujita, 2008). Besides, it is highly speculative (and probably wrong) to say that all glaciers in HMA*

*are "mainly" of this type. See the classifications by Rupper and Roe (2008) or the classification by Maussion et al., (2014), which shows that large parts of HMA are not of the summer accumulation type.*

**Reply:** Yes. In contrast to glaciers in higher latitudes, many on the Tibetan Plateau are summer accumulation type (e.g. Fujita et al., 2000), that is both surface snow fall and melting occur overwhelmingly in the 3 summer months of June, July and August, with little mass gain or loss throughout the remaining 9 months of the year. However some glaciers, especially in the northwestern parts of HMA are winter accumulation type (Maussion et al., 2014).

*L85: I don't understand the need to use different inventories in this study. It seems much more consistent to stick to one, and give all the figures for the one judged more adapted.*

**Reply:** We only use Randolph Glacier Inventory (RGI) 5.0 for glacier outlines. But different parts of the region uses different sources. The RGI 5.0 data inside China are based on the Second Chinese Glacier Inventory (Guo et al., 2015), which provides glacier outlines from 2006–2010, except for some older outlines from the First Chinese Glacier Inventory where suitable imagery could not be found - mainly in southern and eastern Tibet (the S and E Tibet RGI 5.0 sub-region), most of which were made in the 1970s. The RGI 5.0 data outside China are from the "Glacier Area Mapping for Discharge from the Asian Mountains" (GAMDAM) inventory (Nuimura et al., 2015) and nearly all come from 1999–2003 with images selected as close to the year 2000 as possible.

*L90: please justify your choice of the median for the ELA proxy. What consequences does this choice have in the case of glaciers which are far from equilibrium, as it is the case in Eastern Himalaya?*

**Reply:** In Section 3.3 In choosing the initial ELAs for each glacier, there are several reasonable alternatives (Zhao et al., 2016): i) using ELAs interpolated from the first Chinese glacier inventory, ii) median elevations from RGI dataset, iii) the elevation of the 60th percentile of the cumulative area above the glacier terminus. These three choices lead to a range of about 2.5 mm of global sea level in glacier volume loss at 2050. In this study, we use median elevations from RGI dataset, which corresponds to the median result

Table 2 in the revision shows that our model indicates E. Himalaya is in the largest negative mass balance of the sub-regions, in agreement with Gardner et al., 2013.

| Sub-regions | Gardner and others (2013) | Modelled |
|---|---|---|
| E Himalaya | -0.89±0.18 | -1.51±0.59 |

*L99-100: rephrase*

**Reply:** done.

*Table 1: explain the gradients column in the legend, specify units*

**Reply:** The unit of SMB gradients is m m$^{-1}$. We add it in the legend.

*L120: reformulate "to calculate two or three SMB gradients with altitude", which is unclear to me*
**Reply:** We change it to "We calculate no more than three SMB gradients using in-situ SMB measurements for every glacier in Fig.1 and Table 1. Following Zhao et al (2014), the SMB–altitude profile is constructed for every glacier by using its own ELA and these SMB gradients."

*L125: volume area scaling must be extended with a relaxation time scale! See Marzeion et al., (2012) and Bahr et al., (2015).*
**Reply:** We add relaxation time scale in the volume-area scaling, the same as in Marzeion et al., (2012).

*L127: "by assuming all the decrease in area takes place in the lowest parts of the glacier": but how do you deal with growing glaciers?*
**Reply:** We add how to deal with growing glaciers in the revision in Section 3.1 and see the answer to the first main point of the referee.

*L143: "relatively small (<10%).": I wonder as to which percentage the authors would consider that the precipitiation changes aren't "relatively small" anymore. I personally find that 10% is quite a big deal.*
**Reply:** We considered precipitation in the revision and removed these words.

*L150: why not considering CRU (https://cr udata.uea.ac.uk/cru/data/hrg/), which has a resolution of 0.5deg?*
**Reply:** We used CRU temperature data instead of Berkeley Earth Project in the revision, but compare the two results together in Section 3.3 That simulation was done using temperature alone as the glacier driver, so precipitation for each glacier was constant over time. The simulated climate ensemble mean forced volume losses in the period 2010-2069 were +4% (G3), -9% (G4), -11% (RCP4.5) and -13% (RCP8.5) different from the results using the CRU dataset.

*L166: how are they different?*
**Reply:** Yu et al. (2015), noted that was no significant change in surface temperatures after sulphate was injected in the GISS-E2-R model possibly due to the efficacy of SO2 forcing being relatively small as compared to CO2 forcing in the model. Neither do we also find a termination effect in GISS-E2-R under G3. Therefore, we not use any results from GISS-E2-R.

*At the end of the methods section the reader is left with many questions about how the calibration of the α parameter is done, and how the uncertainties are handled in the study.*
**Reply:** In Section 3.3 we discuss the calibration. For the ELA sensitivity to summer

mean temperature and annual precipitation, we use the zonal mean values from energy-balance modelling of glaciers in HMA by Rupper and Roe (2008). Alternatively, it can be estimated using an empirical formula for ablation and a degree-day method (Zhao et al., 2016). Zhao et al. (2016) calculated the ELA for nine glaciers in China, India and Kyrgyzstan, and compared them with the observed ELA time series by similarities of decadal trends and also annual variability. The Rupper and Roe ELA parameterization produced the best fits to observed ELA decadal trends on 9 glaciers, with a correlation coefficient of 0.6 which is significant (p<0.05, the values we give for p are single tailed Pearson correlation tests).

*Fig 2 Fig 3: please make a figure following today's standards. Add country borders or topography (or anything that helps for orientation). Consider using discrete levels instead of continuous colors. Are the anomalies for the entire year or just the summer season?*
**Reply:** We add country borders in Figure 1. As suggested by the other referee, we replaced Fig 2 and 3 with sub-regional line plot plots (the new Fig. 3 in the revision).

*Fig 5: add the spread between the ensemble members*
**Reply:** done.

*Fig 6: the uncertainty associated with the various ensemble members should also appear in the spread.*
**Reply:** done.

*L317: deep convection*
**Reply:** Changed.

*Conclusions: part of the conclusions should be extended and moved to the discussion (in particular the comparison with other studies).*
**Reply:** done.

*L368: specify what "close" means*
**Reply:** We delete "close" and write "The results projected by our method have higher means but smaller uncertainties than theirs, but do not differ significantly."

*References*
*Bahr, D. B., Pfeffer, W. T. and Kaser, G.: A review of volume-area scaling of glaciers, Rev. Geophys., 95–140, doi:10.1002/2014RG000470, 2015.*
*Fujita, K. and Ageta, Y.: Effect of summer accumulation on glacier mass balance on the Tibetan Plateau revealed by mass-balance model, J. Glaciol., 46(153), 244–252, doi:10.3189/172756500781832945, 2000.*
*Gardner, A. S., Moholdt, G., Cogley, J. G., Wouters, B., Arendt, A. a, Wahr, J., Berthier, E., Hock, R., Pfeffer, W. T., Kaser, G., Ligtenberg, S. R. M., Bolch, T., Sharp, M. J., Hagen, J. O., van den Broeke, M. R. and Paul, F.: A Reconciled Estimate of Glacier*

*Contributions to Sea Level Rise: 2003 to 2009, Science., 340(6134), 852–857, doi:10.1126/science.1234532, 2013.*

*Marzeion, B., Jarosch, a. H. and Hofer, M.: Past and future sea-level change from the surface mass balance of glaciers, Cr yosph., 6(6), 1295–1322, doi:10.5194/tc-6-1295-2012, 2012*

*Maussion, F., Scherer, D., Mölg, T., Collier, E., Cur io, J. and Finkelnburg, R.: Precipitation Seasonality and Variability over the Tibetan Plateau as Resolved by the High Asia Reanalysis\*, J. Clim., 27(5), 1910–1927, doi:10.1175/JCLI-D-13-00282.1, 2014.*

*Michaelsen, J.: Cross-validation in statistical climate forecast models, J. Clim. Appl. Meteorol.,26(11),1589–1600,doi:10.1175/1520-0450(1987)026 <1589:CVISCF> 2.0. CO; 2, 1987.*

*Rupper, S. and Roe, G.: Glacier Changes and Regional Climate: A Mass and Energy Balance Approach, J. Clim., 21(20), 5384–5401, doi:10.1175/2008JCLI2219.1, 2008.*

*Interactive comment on Atmos. Chem. Phys. Discuss., doi:10.5194/acp-2016-830, 2016.*

---

## Author Response (AR2)

In the reply, the referee's comments are in *italics*, our response is in normal text, and quotes from the manuscript are in blue.

Suggestions for revision or reasons for rejection (will be published if the paper is accepted for final publication)

*The authors have addressed my comments very well. However, I have a number of concerns about new material in the paper.*

**General comments**

*An altitude precipitation lapse-rate of 3%/100m is introduced with no explanation, no justification and no citations. I'm unfamiliar with this term and a quick literature search didn't enlighten me. If this is to be used it needs to be properly explained, justified and cited for those unfamiliar with this approach to be convinced of its validity.*

**Reply:** The altitude precipitation lapse-rate of 3%/100m is from Marzeion et al. (2012). We add this citation in the place.

*It is great that the authors have included precipitation in their analysis but I was still left somewhat in the dark about what difference it had made. I suggest two changes to address this:*

*1) The old results are presented at the end of section 3 but it seems as if this ought to have been left for section 5 where a quantitative comparison of the two approaches could have been usefully presented.*

**Reply:** Yes. We move the old results to section 5.2 and do the quantitative comparison there in the revision.

*2) It would be great to have the role of precipitation change explicitly addressed in the abstract or conclusion. A paragraph buried in the middle of section 5.1 explains that the widely reported reduction in mean precipitation expected for solar geoengineering is unlikely to be as important as the temperature-driven shift from solid to liquid precip for forcing Himalayan glacier change. It would be great to see this important piece of context brought out more prominently.*

**Reply:** Yes. We add " The widely reported reduction in mean precipitation expected for solar geoengineering is unlikely to be as important as the temperature-driven shift from solid to liquid precipitation for forcing Himalayan glacier change." in the abstract.

*HadGEM2-ES' G4 run is excluded from the glacier analysis as the mass loss is far lower than the other models. This is not a principled reason to exclude it and one should be given or else HadGEM2-ES' G4 experiment should be retained. Such a reason ought to be formulated in terms of the properties of the model, its implementation of the G4 experiment, or its climate features. This reason should also be consistent with keeping the G3 run from this model.*

**Reply:** Agreed, we now include all the models in the means quoted in the paper. We also provide an explanation on the result:

Volume loss using the climate projected by HadGEM2-ES under G4 is far less than that by other models (Table 5), this produces a larger standard deviation for the results than for other scenarios in Table 5. The cause is the combination of small precipitation decrease under RCP4.5 and the G4 anomaly, accompanied by only modest warming (Table 2).

*The anomalous rapid growth of Tibetan glaciers in experiment G3 and in the initial years of G4 is not explained. Is it the case that this model predicts the rapid growth of these glaciers under current conditions? If so, is that consistent with observations?*

**Reply:** Yes. We explained some possible reasons in the last paragraph of section 4.2.2, and add more words to talk about this in the revision. Furthermore, the observations offer some support to the model simulations. Liu et al. (2006) and Shi et al. (2006) found that over 40% of the glaciers in Gangrigabu Mountain in S and E Tibetan Plateau were advancing since the mid-1980s, which is a peculiar phenomena and due to the increase of high precipitation brought by India monsoon.

The rapid growth of glaciers in the S and E Tibet subregion in the initial years of G3 and G4 (Fig. 7) is predict under RCP4.5 scenarios projections with model bias correction for the period 2014-2019 and the G3 or G4 projections for the period after 2020.

Then it would not be surprising that more glaciers advance and there is an increasing trend of glaciers volume in S and E Tibet under the RCP4.5 and G3, G4 scenarios. But the magnitude of volume increase is possibly enlarged because the glacier data inside S and E Tibet was measured in 1970s (the only subregion with old data) and contains outlines of glacier complexes rather than individual glaciers, which has an impact on the volume estimate because of the non-linearity of volume-area scaling relationship.

*L58 – "glacier responses to geoengineering scenarios has been limited to…" The start of this sentence needs some work.*

**Reply:** we change it to

The impact of geoengineering scenarios on ice sheets and glaciers has been limited to studies on global responses …

*L127 – This explanation of the terms could be clearer; I'd suggest either using a list or the "respectively" form rather than a combination of both.*

**Reply:** OK. We change " $V(n+1)$ , $dA(n+1)$ are glacier volume and area change rate

in the $n+1_{th}$ year, respectively" to " $V(n+1)$ is glacier volume and $dA(n+1)$ is area

change rate in the $n+1_{th}$ year".

L177 – The phrasing of this is a little off. The design of experiment G3 should be explained. It is no accident that there is no net change in radiative forcing, that was part of the experimental design.
Reply: "In the 50 years of geoengineering under G3 there is close to a balance between reduction of incoming shortwave radiation and the increase in greenhouse gas forcing" is changed to " In the 50 years of geoengineering, G3 is designed to achieve a balance between reduction of incoming shortwave radiation and the increase in greenhouse gas forcing".

L181 – I would have liked to see a table of the global-mean or Himalayan-mean temperature change for each model and for each experiment here. This would be a useful reference point for some of the comparisons discussed.
**Reply:** Done, we added Table 2 in the revision, showing temperature and precipitation anomalies over 2030-2069 for each model and scenario in the glaciated region of High Mountain Asia relative to their RCP4.5 2010-2029 values.

Eqns 7 and 8 – what does the letter i refer to and what are the listed numbers in reference to?
**Reply:** the letter i refer to the $i$th month in the year. i=6,7,8 means June, July and August.

L252 – "These three choices lead to a range of about 2.5mm of global sea level in glacier volume loss at 2050." What is the context for this figure?
**Reply:** The context is that Zhao et al. (2016) found this result. We added the reference to the sentence.

L275-278 – "…with data regional…" other way around?
**Reply:** Yes, we change "…with data regional…" to "…with regional data…".

Figure 2 and others – "downscaled AND BIAS-CORRECTED"
**Reply:** we add "and bias-corrected".

Figure 3 – The axis labels on this figure need to be changed to be clearer how different panel C and D are. Panel D shows only solid precipitation and not precipitation as a casual glance would suggest. In addition, is the "across model spread" the full range or is it the standard deviation?
**Reply:** Yes, we change the axis label in panel D to "solid precipitation". The "across model spread" is the full range.

L390 and throughout – how are these rates calculated? Are these mean rates, peak rates or rates for a specific decade?
**Reply:** The rate in L390 are annual mean volume loss rates (referenced to the volume in the year 2010) in the first 15 years of post-geoengineering (2070-2084). "There is a clear increase in volume loss rate under G3 after 2069 when geoengineering is terminated. Comparing the last 15 years of geoengineering (2055-2069) with the first 15 years of post-geoengineering (2070-2084) shows annual mean volume loss rate for all the glaciers of 0.17% $a^{-1}$ (referenced to the volume in the year 2010) increases to 1.11% $a^{-1}$, which is higher than the annual mean volume loss rates of 0.54% $a^{-1}$ for RCP4.5 and 0.66% $a^{-1}$ for RCP8.5 in the period 2070-2084."

L530 – "11.1, 12.5…." – add comma
**Reply:** done

L559 – Your results do not have "smaller uncertainties" as yours are not a full measure of uncertainty.
**Reply:** we remove the word "but smaller uncertainties".

L584 – add "respectively" following the glacier retreat figures or else rephrase the sentence.
**Reply:** we add "respectively" following the glacier retreat figures.

[revised manuscript text omitted]

医

Table 2 Summer mean temperature ($\Delta T$) and annual solid precipitation ($\Delta P$) anomalies over 2030-2069 for each model and scenario in the glaciated region of High Mountain Asia relative to their RCP4.5 2010-2029 values.

| Model \ Scenarios | $\Delta T$ (°C) | | | | $\Delta P$ (mm yr⁻¹) | | | |
| --- | --- | --- | --- | --- | --- | --- | --- | --- |
| | G3 | G4 | RCP4.5 | RCP8.5 | G3 | G4 | RCP4.5 | RCP8.5 |
| BNU-ESM | -0.26 | -0.15 | 1.06 | 1.77 | 3.6 | 9.4 | -53.6 | -84.7 |
| CanESM2 | | 0.67 | 1.48 | 2.27 | | -29.2 | -19.6 | -43.0 |
| HadGEM2-ES | 0.48 | 0.10 | 1.09 | 1.71 | -10.3 | 0.2 | -19.0 | -45.7 |
| IPSL-CM5A-LR | 0.32 | | 1.44 | 2.18 | -16.8 | | -50.1 | -72.0 |
| MIROC-ESM | | 0.76 | 1.30 | 1.99 | | -29.2 | -51.6 | -64.2 |
| MIROC-ESM-CHEM | | 0.97 | 1.29 | 2.23 | | -19.6 | -30.3 | -66.5 |

|  |  |  |
| --- | --- | --- |
|  |  |  |
|  |  |  |
|  |  |  |
|  |  |  |
|  |  |  |
|  |  |  |
|  |  |  |

Table 3 climate models and datasets used in this study.

| Name | Reference | Resolution | Data sets |
| --- | --- | --- | --- |
| CRU | Harris et al., 2014 | 0.5°× 0.5° | Surface temperature 1980-2013 |
| GPCC | Becker et al., 2013 | 0.5°× 0.5° | Precipitation 1980-2013 |
| BNU-ESM | Ji et al., 2014 | 2.8°× 2.8° | G3,G4, RCP4.5, RCP8.5 |
| CanESM2 | Arora et al., 2011 | 2.8°× 2.8° | G4, RCP4.5, RCP8.5 |
| HadGEM2-ES | Collins et al., 2011 | 1°× 1.9° | G3,G4, RCP4.5, RCP8.5 |
| IPSL-CM5A-LR | Dufresne et al., 2013 | 1.9°× 3.8° | G3, RCP4.5, RCP8.5 |
| MIROC-ESM | Watanabe et al., 2011 | 2.8°× 2.8° | G4, RCP4.5, RCP8.5 |
| MIROC-ESM-CHEM | Watanabe et al., 2011 | 2.8°× 2.8° | G4, RCP4.5, RCP8.5 |

Table 4 The average rate of elevation change (m a⁻¹) for all the glaciers in sub-

880 regions compared with remote-sensing estimates from 2003 to 2009 from Gardner and others (2013).

| Sub-regions | Gardner and others (2013) | Modelled |
|---|---|---|
| Hissar Alay and Pamir | -0.13±0.22 | -0.02±0.49 |
| S and E Tibet | -0.30±0.13 | -0.39±0.75 |
| Hindu Kush and Karakoram | -0.12±0.15 | -0.08±0.29 |
| W Himalaya | -0.53±0.13 | 0.32±0.29 |
| C Himalaya | -0.44±0.20 | -0.62±0.63 |
| E Himalaya | -0.89±0.18 | -1.51±0.59 |
| All HMA | -0.27±0.17 | -0.13±0.60 |

Table 5. The volume loss in mm sea-level equivalent, projected using forcing from all the climate models in the period 2010-2069 and 2070-2089 post-geoengineering period under G3, G4, RCP4.5 and RCP8.5. The means of volumes lost driven by individual model forcing and its standard deviation are shown in the penultimate row. The simulated volume loss using the climate model ensemble mean forcing of temperature and precipitation is shown in the last row. The volume loss is calculated by assuming ice density of 900 kg m$^{-3}$ and ocean area of $362\times10^{12}$ m$^2$.

890

| Scenarios | G3 | | G4 | | RCP4.5 | | RCP8.5 | |
|---|---|---|---|---|---|---|---|---|
| Period / Model | 2010-69 | 2070-89 | 2010-69 | 2070-89 | 2010-69 | 2070-89 | 2010-69 | 2070-89 |
| BNU-ESM | 10.2 | 5.3 | 11.0 | 5.5 | 18.5 | 2.5 | 20.8 | 3.2 |
| CanESM2 | ---- | ---- | 8.3 | 4.1 | 14.0 | 2.0 | 17.8 | 3.5 |
| HadGEM2-ES | 7.2 | 3.4 | 3.2[±] | 3.7[±] | 12.0 | 2.5 | 15.9 | 4.7 |
| IPSL-CM5A-LR | 9.8 | 6.3 | ----- | ---- | 16.7 | 3.2 | 19.5 | 3.8 |
| MIROC-ESM | ----- | ----- | 12.6 | 4.0 | 15.8 | 3.0 | 19.0 | 3.9 |
| MIROC-ESM-CHEM | ----- | ----- | 14.0 | 3.8 | 16.0 | 2.9 | 19.1 | 3.1 |
| Mean ± std | 9.0±1.6 | 5.4±1.0 | 9.8±4.3 | 4.2 ± 0.7 | 15.5±2.3 | 2.7±0.4 | 18.5±1.7 | 3.7±0.6 |
| Ensemble mean climate forcing | 8.1 | 5.9 | 11.7 | 4.7 | 16.6 | 2.9 | 19.2 | 3.6 |